# Malignant subclone drives metastasis of genetically and phenotypically heterogenous cell clusters through fibrotic niche generation

Sau Yee Kok[1], Hiroko Oshima[1,2], Kei Takahashi[3], Mizuho Nakayama[1,2], Kazuhiro Murakami[4], Hiroki R. Ueda[5,6], Kohei Miyazono [3] & Masanobu Oshima [1,2✉]

A concept of polyclonal metastasis has recently been proposed, wherein tumor cell clusters break off from the primary site and are disseminated. However, the involvement of driver mutations in such polyclonal mechanism is not fully understood. Here, we show that non-metastatic AP cells metastasize to the liver with metastatic AKTP cells after co-transplantation to the spleen. Furthermore, AKTP cell depletion after the development of metastases results in the continuous proliferation of the remaining AP cells, indicating a role of AKTP cells in the early step of polyclonal metastasis. Importantly, AKTP cells, but not AP cells, induce fibrotic niche generation when arrested in the sinusoid, and such fibrotic microenvironment promotes the colonization of AP cells. These results indicate that non-metastatic cells can metastasize via the polyclonal metastasis mechanism using the fibrotic niche induced by malignant cells. Thus, targeting the fibrotic niche is an effective strategy for halting polyclonal metastasis.

[1] Division of Genetics, Cancer Research Institute, Kanazawa University, Kanazawa, Japan. [2] WPI Nano Life Science Institute, Kanazawa University, Kanazawa, Japan. [3] Department of Molecular Pathology, Graduate School of Medicine, The University of Tokyo, Tokyo, Japan. [4] Division of Epithelial Stem Cell Biology, Cancer Research Institute, Kanazawa University, Kanazawa, Japan. [5] Department of Systems Pharmacology, The University of Tokyo, Tokyo, Japan. [6] Laboratory for Synthetic Biology, RIKEN BDR, Suita, Osaka, Japan. ✉email: oshimam@staff.kanazawa-u.ac.jp

Colorectal cancer (CRC) is a leading cause of cancer-related death globally[1,2], and the majority of cancer-related deaths are caused by metastasis[3]. It is therefore important to understand the molecular and cellular mechanisms underlying metastasis in order to establish a clinical strategy to overcome cancer mortality[4].

The accumulation of driver mutations is understood to be responsible for the malignant progression of CRC[5], and genome research has identified driver genes that are frequently mutated in CRC cells and are suspected of promoting each step of malignant progression[6,7]. Using genetic mouse models, we previously examined the link between specific combinations of driver mutations and each process of intestinal tumor progression, including submucosal invasion, epithelial mesenchymal transition (EMT)-like morphology, intravasation, and metastasis[8–11]. However, recent bioinformatics analyses have indicated that most driver mutations are accumulated in primary CRCs, suggesting that the metastasis process is not promoted by genetic alterations and subsequent clonal selection[12,13].

Recently, a concept polyclonal metastasis has been proposed[14–17]. In this model, tumor cell clusters are detached from the primary site, circulate in the blood stream as circulating tumor cell (CTC)-clusters, disseminate, and colonize in the distant organs[16]. It has also been reported that increased numbers of CTC-clusters are associated with a worse prognosis in breast cancer patients[18]. This polyclonal metastasis mechanism can explain how genetic diversity is transferred from the primary site to metastatic lesions[19]. However, most studies that contributed to the polyclonal metastasis concept used differentially labeled but genetically identical or similar cancer cells, so how genetically heterogenous cells in the primary tumors are selected for cluster formation and colonization in the distant organs by the polyclonal mechanism remains unclear.

To answer this question, we performed spleen transplantation experiments using a recently established organoid transplantation model[8,11]. In this model, mouse intestinal tumor-derived organoids carry various combinations of genetic alterations in driver genes. We demonstrate that non-metastatic cells can metastasize when co-disseminated with metastatic cells through the generation of a fibrotic microenvironment in the arrested liver vessels.

## Results

### A metastasis model using genetically defined tumor organoids.
We previously constructed compound mutant mice that carried $Apc^{\Delta716}$ (A), $Kras^{G12D}$ (K), $Tgfbr2^{-/-}$ (T), and $Trp53^{R270H}$ (P) mutations in various combinations and established intestinal tumor-derived organoids with different genotypes[8–10]. The organoid cells used in the present study are A, AK, AT, AP, and AKTP. A and AK are non-invasive non-metastatic cells, while AT and AP are invasive non-metastatic cells (Supplementary Fig. 1). In contrast, AKTP cells are invasive metastatic cells.

### Metastases of non-metastatic cells with metastatic cells.
In the polyclonal metastasis model, tumor cell colonization in the distant organs originates from disseminated cell clusters[14,20]. We therefore examined whether liver metastatic foci in the spleen transplantation model originate from cell clusters or single cells. To assess this, differentially labeled AKTP cells with Venus or tdTomato were mixed and transplanted to the spleen, and liver tissues were examined chronologically (Fig. 1a). At day 3 after spleen transplantation, tumor cell clusters including >10 cells were frequently found inside the sinusoid vessels of the liver, and the majority of tumor clusters consisted of a mixture of Venus- and tdTomato-labeled cells (Fig. 1b, c). Considering the duration of time of 3 days and cell numbers and mixed colors in each

cluster, these cells were thought to have been arrested as clusters rather than proliferated from single cells.

At days 7, 14, and 28 after transplantation, multiple tumor foci had developed, consisting of a mixture of Venus- and tdTomato-labeled cells or either of Venus- or tdTomato-labeled cells at similar ratios throughout the examined time points (Fig. 1c, d), indicating that the majority of metastatic foci in the liver in this model originate from cell clusters. We further examined the liver tissues at seven weeks after transplantation using a tissue-clearing protocol called a clear, unobstructed brain/body imaging cocktails and computational (CUBIC) analysis[21]. Notably, multiple metastatic foci developed, and yellow-colored tumor lesions were found in the merged images (Fig. 1e, arrowheads). Although the chimeric metastasis lesions were difficult to distinguish from merged adjacent monoclonal metastases on CUBIC images, we confirmed the mixture of Venus- and tdTomato-labeled tumor cells in the metastatic lesions by immunohistochemistry (Fig. 1f). We therefore used this model for the analysis of polyclonal metastasis.

It has been shown that a malignant subclone can accelerate metastasis of cell clusters that include less metastatic cells[22], suggesting that non-metastatic cells can metastasize by forming cell clusters with malignant cells. To examine this possibility, we co-transplanted Venus-labeled AKTP cells with tdTomato-labeled non-metastatic A, AK, AT, or AP cells to the spleen at a ratio of 1:1, and metastasized liver tumors were examined at 4 weeks after transplantation (Fig. 2a). We first confirmed that A, AK, AT, and AP cells did not metastasize to the liver when they were transplanted to the spleen, while AKTP cells formed multiple metastatic foci (Fig. 2b). When non-invasive and non-metastatic A and AK cells were co-transplanted with AKTP cells, all of the metastatic lesions in the liver were green in color, indicating that A and AK cells failed to metastasize, even if metastatic AKTP cells co-existed in the primary sites (Fig. 2c, d left). Importantly, when invasive but non-metastatic AT and AP cells were co-transplanted with AKTP cells, subpopulations of less than 20% of liver metastatic foci showed dual green and red colors (Fig. 2c), and a mixture of Venus- and tdTomato-labeled cells in the metastatic tumors was confirmed by immunohistochemistry (Fig. 2d right). Furthermore, immunohistochemistry for Ki67 indicated that the proliferation rate of tdTomato-labeled AT and AP cells in the metastatic foci was similar to that of AKTP cells (Supplementary Fig. 2). The ratios of AT and AP cells in each of the metastatic foci varied; however, none of tumor lesions contained only AT or AP cells, with the exception of one small lesion that contained only AP cells (Fig. 2e). These results indicate that non-metastatic AT and AP cells can metastasize only when metastatic AKTP cells coexist in the primary sites, although the efficiency of chimeric metastases is significantly lower than that of AKTP cell monoclonal metastases (Fig. 2c, e). Because A and AK cells cannot metastasize with AKTP cells, it is possible that the invasive ability of tumor cells is a minimum prerequisite for metastasis by the polyclonal mechanism.

### Requirement of co-dissemination for polyclonal metastasis.
It has been suggested that heterogeneity in metastatic lesions can also be generated by serial seeding of CTCs to established metastatic tumors[17]. To test this possibility, Venus-labeled AKTP cells were transplanted to the spleen, and two weeks later, tdTomato-labeled cells were directly injected into the portal vein to avoid possible co-dissemination with Venus-AKTP cells from the original transplanted site (Supplementary Fig. 3a). When tdTomato-labeled AKTP cells were injected via the portal vein, we found liver metastases consisting of both tdTomato- and Venus-AKTP cells as well as only tdTomato-AKTP cells

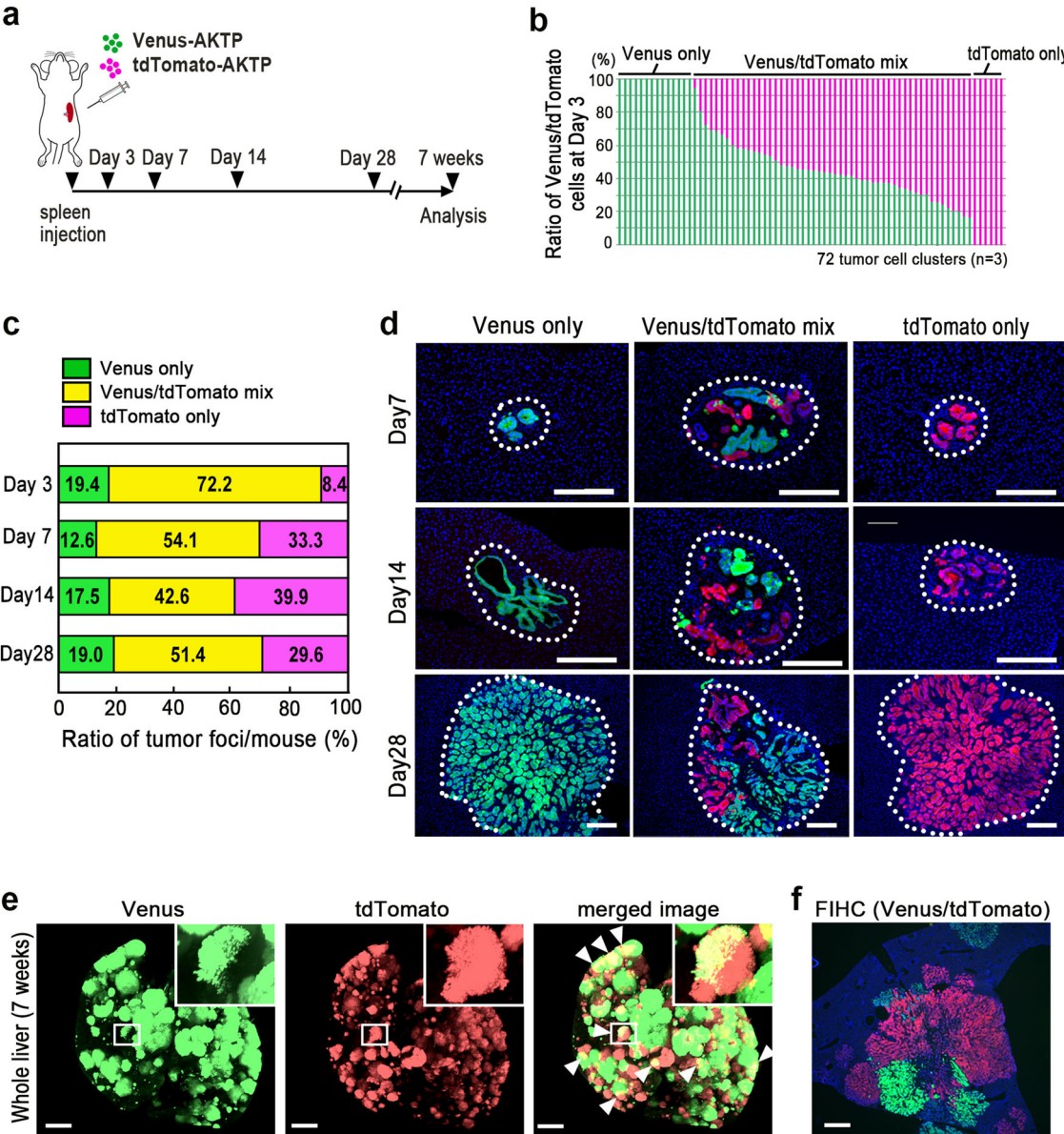

**Fig. 1 The polyclonal origin of liver metastases in the spleen transplantation model. a** Schematic illustration of the transplantation experiment. Venus-labeled AKTP cells (green) and tdTomato-labeled AKTP cells (red) were mixed and transplanted to the spleen. Liver tissues were examined at the indicated time points. **b** The ratios of Venus-AKTP (green bars) and tdTomato-AKTP cells (magenta bars) in 72 tumor cell clusters developed in $n = 3$ biologically independent mice at day 3 are indicated. **c** The ratios of metaplastic foci consisting of Venus-AKTP only (green), tdTomato-AKTP only (magenta), and a mixture of Venus- and tdTomato-AKTP cells (yellow) at each time point are indicated as a bar graph. **d** Representative photographs of the fluorescent immunohistochemistry of tumor lesions consisting of Venus-AKTP only (left, green), tdTomato-AKTP only (right, red), and mixed Venus- and tdTomato-AKTP cells (center, green/red) at day 7 (top), day 14 (middle), and day 28 (bottom) after transplantation. Tumor foci are indicated by dotted lines. Bars, 200 μm. **e** Representative photographs of fluorescence imaging of the whole liver (CUBIC images) to detect Venus (488 nm) and tdTomato (590 nm), along with a merged image (left to right). Insets indicate enlarged images of the boxed areas. Arrowheads on the merged image indicate chimeric tumor lesions (yellow). Bars, 4 mm. **f** A representative photograph of the fluorescent immunohistochemistry (FIHC) of Venus- and tdTomato-AKTP cells in the liver at 7 weeks after transplantation. Bar, 1 mm. The photographs in **d**, **e**, and **f** are representative images from $n = 3–4$ independent animals. Source data are provided as a Source Data File.

(Supplementary Fig. 3b top A and B, respectively), suggesting that AKTP cells can generate metastases either via clonal dissemination or serial seeding. When tdTomato-labeled AT or AP cells were injected into the portal vein as a secondary injection, all liver tumors consisted of Venus-labeled AKTP cells alone, and tdTomato-labeled AT and AP cells were not found by the histological examination. These results indicate that AT and AP cells need co-dissemination with AKTP cells for the development of polyclonal metastases rather than serial seeding.

**Growth of non-metastatic cells after metastatic cell depletion.** We next examined whether or not AKTP cells are continuously required for the survival and proliferation of AT and AP cells in the liver after the establishment of polyclonal metastases. To deplete AKTP cells from the chimeric tumor tissues, we constructed AKTP-DTR cells expressing diphtheria toxin receptor (DTR). We confirmed that the treatment of AKTP-DTR cells with diphtheria toxin (DT) caused cell death within 48 h (Supplementary Fig. 4). As a control experiment, Venus-labeled AKTP-DTR cells and

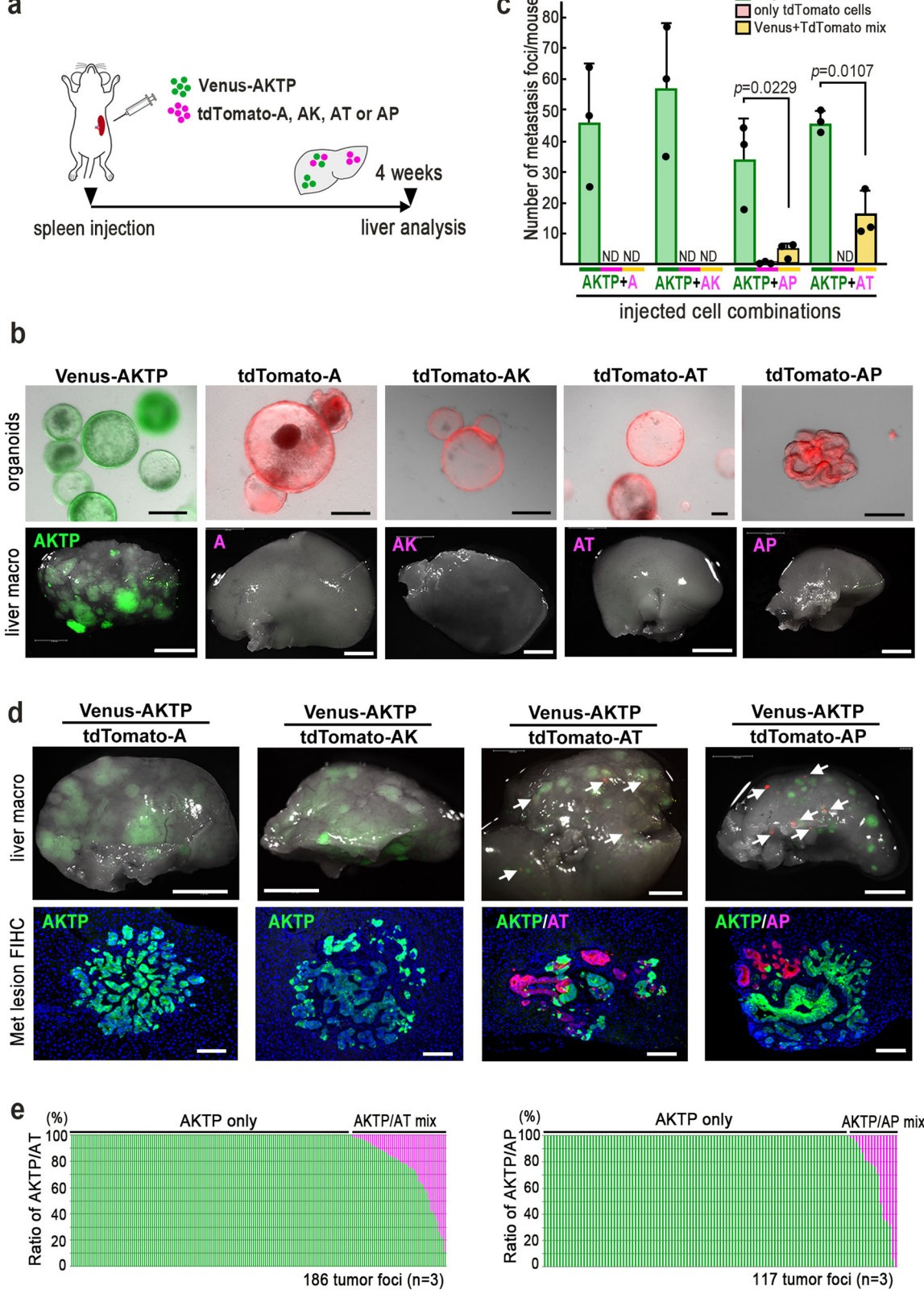

tdTomato-labeled AKTP cells were mixed and co-transplanted to the spleen, and DT was continuously administered to mice for five weeks (Fig. 3a). Liver tissues were then examined by CUBIC imaging. DT treatment successfully reduced the number of Venus-labeled AKTP-DTR cells, while tdTomato-labeled AKTP cells remained and formed multiple metastatic foci (Fig. 3b).

We next co-transplanted Venus-labeled AKTP-DTR cells and tdTomato-labeled AP cells (Fig. 3a). In the non-DT control mice, AKTP-DTR cells were predominantly found in metastatic lesions, and the partial contribution of AP cells was confirmed in a subpopulation of tumor lesions (Fig. 3c, top, arrowheads). In contrast, in the DT-treated mice, most AKTP-DTR cells were

**Fig. 2 Polyclonal metastasis of non-metastatic cells with metastatic cells. a** Schematic illustration of the transplantation experiment. Venus-labeled AKTP cells (green) were transplanted with tdTomato-labeled A, AK, AT, or AP cells (red), and liver tissues were examined at four weeks after transplantation. **b** Representative photographs of organoids (top) and livers of the mice transplanted with the indicated genotype organoids (bottom) under fluorescent dissection microscopy. Bars, 250 μm (top) and 3 mm (bottom). **c** The numbers of metastatic lesions with AKTP cell-only (green), AP cell-only (red), or a mixture of AKTP with AT or AP cells (yellow) on histological sections of mice transplanted with the indicated genotype organoids are indicated as a bar graph (n = 3 biologically independent animals for each experimental condition). Data are presented as the mean ± s.d. A two-sided unpaired t-test was used to calculate the significance of differences, and p values are provided. ND not detected. **d** Representative photographs of livers under a fluorescent dissection microscope (top) and fluorescent immunohistochemistry (FIHC) of metastasis (Met) lesions (bottom) of the mice transplanted with the indicated combination of organoids. Arrows indicate metastatic foci with tdTomato-labeled cells. Bars, 3 mm (top) and 100 μm (bottom). **e** The ratio of Venus-labeled AKTP cells (green) and tdTomato-labeled AT (left) or AP (right) cells (magenta) in 186 and 117 tumor lesions, respectively, developed in n = 3 biologically independent mice for each transplantation combination. The photographs in **b** (top) are representative images of n = 10 independent cultures. The photographs for **b** (bottom) and **d** (top and bottom) are representative images from n = 3 biologically independent animals. Source data are provided as a Source Data File.

depleted, and the remaining AP cells proliferated to form larger tumors (Fig. 3c, bottom). Histologically, the liver tumor lesions of DT-treated mice consisted of only tdTomato-labeled AP cells, and a high Ki67 labeling efficiency was confirmed (Fig. 3d, top). In the non-DT control of Venus-labeled AKTP-DTR and tdTomato-labeled AT cell-transplanted mice, AT cells were confirmed in a subpopulation of liver tumors (Fig. 3e, top). In contrast to AP cells, remaining AT cells after DT treatment showed a relatively low Ki67 staining efficiency and did not develop large tumors like AP cells (Fig. 3d, e, bottom). The tumor size distribution of AP cells increased significantly after DT treatment compared with that of the non-DT control, and the number of large tumors ($5$–$20 \times 10^5$ μm$^2$ on the histology section) was significantly increased (Fig. 3f, top). In contrast, the tumor size distributions of AT cells were not markedly changed after DT treatment (Fig. 3f, bottom). These results indicate that both AP and AT cells can survive in the liver without continuous support of AKTP cells and that proliferation of AP cells actually increased to form large tumors upon the elimination of AKTP cells. It is therefore possible that driver mutations in *APC* and *TP53* are sufficient to allow proliferation in the liver once cells are colonized via the polyclonal mechanism.

**Fibrotic niche for the colonization of disseminated cells**. The above results suggested that AKTP cells play an important role in the early stage of polyclonal metastasis for both AP and AT cells. We therefore performed a chronological examination of liver tissues from the early stages after dissemination (Fig. 4a).

At day 1 after AKTP cell transplantation to the spleen, tumor cell clusters were arrested inside sinusoid vessels with Matrigel (Fig. 4b, top). At day 3 after transplantation, fibroblastic cells started proliferating along the endothelial cell layers. A fibrotic microenvironment surrounding the AKTP cells was then generated by day 14 (Fig. 4b, top, arrowheads). We confirmed that collagen fibers were deposited in stroma of AKTP cell tumors at day 14 by Masson's trichrome and Sirius red staining (Fig. 4c). Notably, α-smooth muscle actin (αSMA)-positive cells were found inside and outside of the AKTP cell-arrested vessel wall (Fig. 4d, left) and were proliferating (Fig. 4d, e). Accordingly, it is possible that the proliferating fibroblast-like cells in the metastatic niche originated from hepatic stellate cells (HSCs). In contrast, such HSC proliferation and fibrotic niche generation were not found in liver vessels when only AP or AT cells were transplanted (Fig. 4b, middle). We confirmed that only Matrigel transplantation did not cause such a fibrotic host response, excluding a possible effect by Matrigel on fibrotic niche generation (Fig. 4b, bottom). These results indicate that AKTP cells, but not AT and AP cells, acquired an ability to induce fibrotic niche generation via the activation of HSCs.

As expected, when Venus-labeled AKTP cells and tdTomato-labeled AP cells were mixed and co-transplanted to the spleen, the fibrotic niche was generated at two weeks, possibly due to AKTP cell-derived factors (Fig. 4f, top, arrowheads), and chimeric tumors consisting of AP and AKTP cells were confirmed at seven weeks (Fig. 4f, bottom). Notably, when AP and AT cells were arrested in the vessels without AKTP cells, the Ki67 labeling indices were significantly lower at day 14 than at day 7 (Supplementary Fig. 5). However, the Ki67 index of AP cells in the chimeric tumors had significantly increased to a level similar to that of AKTP cells at seven weeks after transplantation (Fig. 4g). These results, taken together, strongly suggest that the AKTP cell-associated fibrotic niche is important for the survival and initial colonization of non-metastatic AP cells.

**Metastatic cell-generated fibrotic niche for colonization**. It has been shown that HSCs play a role in the generation of the metastatic niche for liver metastases[23–27]. We therefore isolated immortalized HSCs from *Trp53* mutant mice (Supplementary Fig. 6) and co-transplanted tdTomato-labeled AP cells and Venus-labeled HSCs to the spleen (Fig. 5a). At two weeks after transplantation, we found multiple small HSC-associated AP cell clusters, which were thought to be arrested cell clusters in the vessels (Fig. 5b). However, fibrotic niche generation like in AKTP cell tumors was not found at two weeks. At four weeks after transplantation, only two metastatic lesions were found in five mice transplanted with a mixture of AP cells and HSCs (Supplementary Fig. 7). Venus-labeled HSCs were not found in any of these lesions, suggesting that co-transplanted HSCs did not sufficiently support AP cell metastasis. It is possible that the activation of HSCs by AKTP cells is required to support non-metastatic cell colonization.

We therefore examined the role of the AKTP cell-generated niche in the colonization of AP cells by transplantation experiments using AKTP-DTR cells. At one week after AKTP-DTR cell transplantation, mice were treated with DT to deplete AKTP-DTR cells, and then AP cells were transplanted to the spleen (Fig. 5c). In the liver vessels at one week after DT treatment, we found collagen fiber deposition and αSMA-positive cells along the vessel wall, which were thought to be the remaining tumor niche generated by AKTP-DTR cells (Fig. 5d). Notably, a significant number of AP cell colonies were found in all mouse livers at three to five weeks after AP cell transplantation, and these tumor lesions were associated with increased numbers of αSMA-positive cells (Fig. 5e, f). Furthermore, collagen fiber depositions were colocalized with fibroblastic cells in the tumor stroma, suggesting that αSMA-expressing fibroblastic cells were major sources of collagen fibers in these foci (Fig. 5e). Such AP cell colonies were not found in the liver when AP cells were transplanted without prior niche generation by

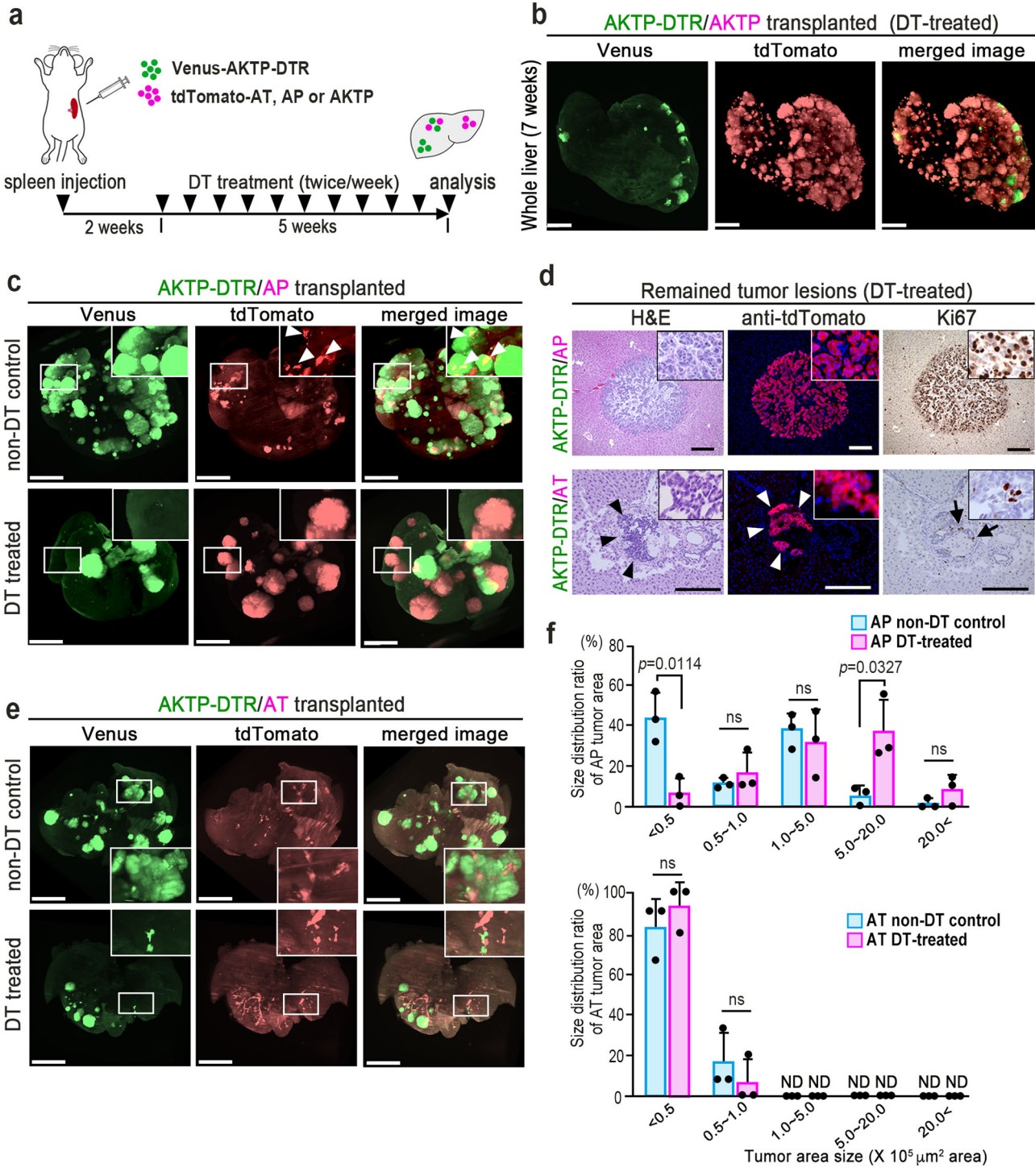

AKTP cells. These results support the idea that the AKTP cell-generated fibrotic niche is required for promotion of AP cell survival and colonization.

**TGF-β for HSC activation and metastatic niche generation.** We therefore stimulated HSCs with conditioned medium (CM) of the respective genotypes of the tumor cell cultures. Notably, stimulation of HSCs with AKTP-CM caused a distinct morphological change to a myofibroblast-like type, suggesting the activation of HSCs[28] (Fig. 6a). AP-CM slightly changed the HSC morphology, while AK-CM and control medium did not induce any

morphological changes. Consistently, the expression of *Tgfb1* and *Mki67* in HSCs was increased significantly upon stimulation with AKTP-CM compared to that with AP- or AK-CM (Fig. 6b), indicating that AKTP cell-secreted factors activate HSCs to generate fibrotic niche.

TGF-β signaling has been shown to play a key role in HSC activation and metastatic niche generation in the liver[23,24]. To examine the functional contribution of activated HSCs in metastasis, we transplanted AKTP cells to the spleen of ROSA26-CreER *Tgfbr2flox/flox* mice, and one week later, *Tgfbr2* encoding TGF-β type II receptor was disrupted by tamoxifen (Tam) treatment (Fig. 6c). Importantly, blocking TGF-β signaling

**Fig. 3 Depletion of metastatic subclone from the polyclonal metastasis lesions. a** Schematic illustration of the transplantation experiment. Venus-labeled AKTP-DTR cells (green) were transplanted with tdTomato-labeled AT, AP, or AKTP cells (red), and mice were treated with diphtheria toxin (DT) twice a week for 5 weeks. Liver tissues were examined at 5 weeks after DT treatment. **b** Representative photographs of fluorescence imaging of the whole liver (CUBIC images) to detect Venus (488 nm, left) and tdTomato (590 nm, center), and a merged image (right) of DT-treated mice transplanted with Venus-AKTP-DTR and tdTomato-AKTP cells. Bars, 4 mm. **c** Representative photographs of fluorescence imaging of the whole liver (CUBIC images) to detect Venus (488 nm, left) and tdTomato (590 nm/532 nm, center), and a merged image (right) of Non-DT control and DT-treated mice transplanted with Venus-AKTP-DTR cells and tdTomato-AP cells. Insets show enlarged images of the boxed areas. Arrowheads indicate AP cells in the mixed tumor foci. Bars, 4 mm. **d** Representative photographs of the liver tumor lesions with remaining AP cells (top) and AT cells (bottom) after DT treatment, H&E staining (left) and immunohistochemistry for tdTomato (center) and Ki67 (right). Insets show enlarged images of the tumor lesions. Arrowheads indicate remaining AT cells. Arrows indicate Ki67-positive AT cells. Bars, 100 μm. **e** Representative photographs of fluorescence imaging of the whole liver (CUBIC images) to detect Venus (488 nm, left) and tdTomato (590 nm, center), and a merged image (right) of the Non-DT control and DT-treated mice transplanted with Venus-AKTP-DTR cells and tdTomato-AT cells. Insets show enlarged images of the boxed areas. Bars, 4 mm. **f** Size distributions of AP (top) and AT cell tumors (bottom) in the non-DT control (blue) and DT-treated mouse livers (magenta) are shown in bar graphs as the % area ($n = 3$ biologically independent animals for each experimental condition). The data are presented as the mean ± s.d. A two-sided unpaired $t$-test was used to calculate the significance of differences, and $p$ values are provided. ns not significant, ND not detected. The photographs in **b**, **c**, **d**, and **e** are representative images from $n = 3$ biologically independent animals. Source data are provided as a Source Data File.

in host cells significantly suppressed the development of metastatic tumors in the liver (Fig. 6d, e). In the small tumor lesions of the *Tgfbr2* knockout mice, the numbers of αSMA-positive cells were not increased in the stroma, and Ki67 staining efficiency of tumor cells was lower than that in No-Tam control mouse tumors (Fig. 6f). These results also suggest that the fibrotic niche originated by AKTP cell-activated HSCs is required for the development of metastatic lesions.

**Interaction of metastatic cells and HSCs for metastasis.** We next co-cultured AKTP cells and HSCs directly to examine the interaction of these two cell types. Interestingly, AKTP cells showed tightly packed embryonic stem (ES) cell-like colonies on the myofibroblast-like HSC cell layer (Fig. 7a). Consistently, a PCR array and RT-PCR analyses indicated that ES cell signature genes, including *Fgf2*, *Ncam1*, *Des*, *Gata6*, *Thy1*, *T*, and *Nkx2-5*, were significantly upregulated in AKTP cells when co-cultured with HSCs (Fig. 7b, Supplementary Fig. 8). In addition, we confirmed that the FGF2 expression was induced in AKTP cells in the liver tumor foci at day 14 but not day 3 after spleen transplantation (Fig. 7c), suggesting a role of HSCs in AKTP cell activation. Accordingly, it is possible that HSCs and AKTP cells interact through a positive feedback mechanism involving mutual activation.

To further examine the effect of AKTP-HSC interaction in metastasis, we stimulated AP cells with AKTP-CM or FGF2, one of upregulated genes in AKTP cells upon co-culture with HSCs. Although the AP cell proliferation rate was not changed by these factors, the cloning efficiency of AP cells was significantly increased by both (Fig. 7d, e), indicating that AKTP-secreting factors including FGF2 promote the clonal expansion of AP cells through a direct effect.

We next cultured AP cells in a non-adherent dish to form spheroids in mono-culture or co-culture with HSCs in the presence or absence of AKTP cell-secreting factors (Fig. 7f, top). In this method, AP cells and HSCs form chimeric spheroids by co-culture. AKTP-CM did not affect the proliferation of mono-cultured AP cells, and co-culture with HSCs in the absence of AKTP-CM also did not change proliferation of AP cells (Fig. 7f, bottom and Fig. 7g). However, the Ki67 index of AP cells in the chimeric spheroids with HSCs was significantly increased upon stimulation with AKTP-CM as well as FGF2. These results indicate that AKTP cell-secreted factors, such as FGF2 indirectly promote AP cell proliferation through the activation of HSCs.

## Discussion

In the present study, we showed that non-metastatic intestinal tumor cells with a low accumulation of driver mutations can metastasize to the liver via a polyclonal mechanism. As a possible mechanism, the malignant metastatic subclone activates HSCs to generate the fibrotic niche in the arrested vessels by secreting certain factors, creating a fibrotic microenvironment that is important for colonization (Fig. 7h, top). In contrast, non-metastatic cells do not sufficiently activate HSCs and thus fail to survive in the liver (Fig. 7h, middle). When non-metastatic cells are co-disseminated with metastatic cells in the vessel, the malignant subclone induces fibrotic niche generation, and non-metastatic cells survive and colonize with the aid of the generated microenvironment (Fig. 7h, bottom). This mechanism of polyclonal metastasis challenges the concept of the multistep model for malignant progression[5], wherein cell clones that acquire metastatic ability through genetic alterations are selected for metastasis. In this study, we found that the interaction of AKTP cells and HSCs is important for the survival and proliferation of AP cells in metastatic lesions. Based on the present results, it is possible that metastatic cells aid proliferation of non-metastatic cells via indirect mechanism through HSC activation by secreting factors including FGF2, while malignant cells may promote cloning efficiency of non-metastatic cells via direct mechanism by their secreting factors.

In this study, we examined the role of HSCs in fibrotic niche generation in liver metastases. However, it has been reported that αSMA-expressing hepatic myofibroblasts also originate from portal fibroblasts or bone marrow-derived fibrocytes[29]. Furthermore, the ratio of different-origin-derived αSMA-expressing cells may change during tumor progression, suggesting the possibility that stromal cells originated from portal fibroblasts or fibrocytes are also recruited in the metastatic foci and contribute to colonization like HSC-originating fibroblast-like cells. However, further investigations will need to be conducted to confirm this point.

In addition, we showed that once polyclonal metastasis is established, AP cells no longer require AKTP cells for continuous proliferation. Therefore, whether or not activated fibroblastic cells play a role at later proliferation stages of polyclonal metastasis will need to be further explored.

Interestingly, we found that not all non-metastatic cells can metastasize by a polyclonal mechanism. A minimum invasive ability in the primary tumors was required for the involvement of polyclonal metastasis in the present study, although the requirement of genetic alterations may depend on the phenotypes of the

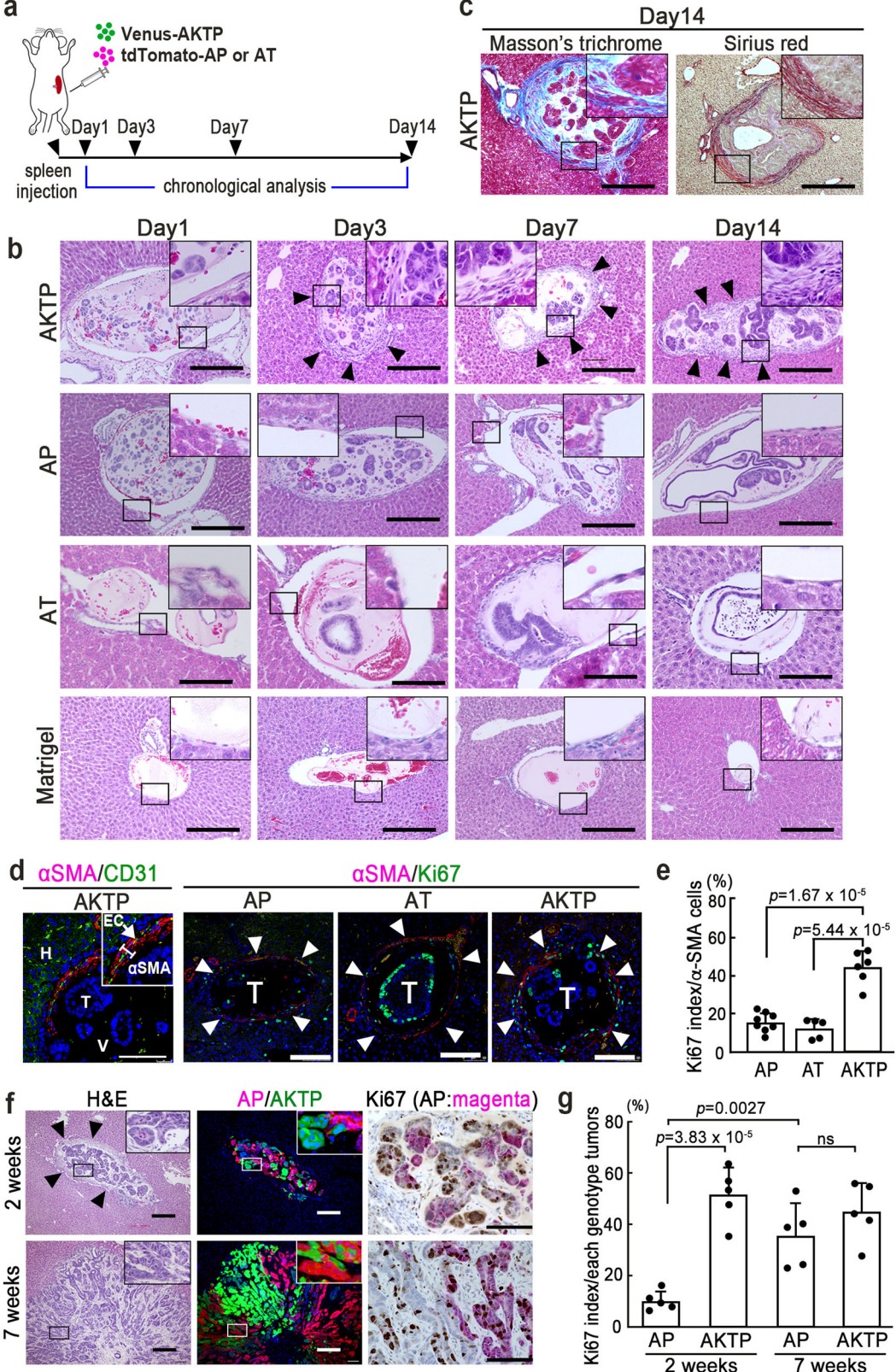

malignant driver subclone. However, if non-metastatic cells are continuously less malignant in liver metastatic tumors than driver subclone cells, the polyclonal mechanism is not important for the clinical strategy. However, our study showed that AP cells with only two gene mutations in *Apc* and *Trp53* were able to survive and continued to proliferate to form large tumors in the liver after

the depletion of AKTP cells (Fig. 3c). Compound mutant mice carrying *Apc* and *Trp53* did not develop metastases of intestinal tumors[10]; thus, AP tumors have been considered non-life-threatening. This point is particularly important to consider when devising a therapeutic strategy. Molecular-targeting drugs against activated KRAS can eliminate malignant cells that carry

**Fig. 4 Fibrotic niche generation surrounding metastatic cells in liver vessels. a** Scheme of the transplantation experiment. Venus-labeled AKTP (green), tdTomato-labeled AP, or AT cells (red) were transplanted, and liver tissues were examined at the indicated days. **b** Representative photographs of tumor cell-arrested liver vessels at the indicated days after spleen transplantation with AKTP (top), AP (middle top), AT cells (middle bottom), or Matrigel (bottom). Insets indicate enlarged images of boxed areas. Arrowheads indicate fibrotic niche. Bars, 100 μm. **c** Representative photographs of Masson's trichrome (left) and Sirius red staining (right) of AKTP cell tumor lesions at day 14 after transplantation. Insets indicate enlarged images of the boxed areas. Bars, 100 μm. **d** The fluorescent immunohistochemistry for αSMA (red)/CD31 (green) in the AKTP cell-arrested liver vessel (left) and αSMA (red)/Ki67 (green) in the AP, AT, or AKTP cell-arrested vessels (right three photographs) at day 14. The arrow indicates endothelial cells, and arrowheads indicate vessel walls. EC endothelial cells, T tumor, V vessel, H hepatocytes. Bars, 50 μm. **e** The Ki67-labeling indices of hepatic stellate cells (HSCs) along the vessels arrested with AP, AT, and AKTP cells are shown as a bar graph ($n = 8$, 5, and 6 biologically independent samples for AP, AT, and AKTP, respectively). **f** Representative photographs of tumor cell clusters at two weeks (top) and seven weeks (bottom) of mice transplanted with Venus-AKTP and tdTomato-AP cells. Representative photographs of H&E (left), fluorescent immunohistochemistry for AP (red)/AKTP cells (green) (center) in serial sections, and immunohistochemistry for Ki67 (brown)/AP cells (magenta) (right). Arrowheads indicate proliferating fibroblast-like cells. Insets show enlarged images of the boxed areas. Bars, 100 μm. **g** The Ki67-labeling indices of AP and AKTP cells in the chimeric tumors at the indicated weeks after spleen transplantation are shown as a bar graph ($n = 5$ biologically independent samples for each experimental condition). The photographs in **b**, **c**, **d**, and **f** are representative images from $n = 3$ biologically independent animals. The data in **e** and **g** are presented as mean ± s.d. A two-sided unpaired $t$-test was used to calculate statistical significance, and $p$ values are provided. ns not significant. Source data are provided as a Source Data File.

*KRAS* mutations (like AKTP cells); however, if non-KRAS mutant cells (like AP cells) co-exist as a minor population in metastatic lesions, they continue to proliferate after the elimination of KRAS mutant cells and form drug-resistant tumors, which may unexpectedly worsen the prognosis. Thus, it is important to understand the genetic heterogeneity of metastatic liver lesions of human CRC at the single-cell level in order to consider an effective therapeutic strategy. However, in the present study, we used a spleen transplantation model to examine the liver metastasis process via portal vein circulation. It thus remains unclear whether or not non-metastatic cells can detach from the primary tumors and intravasate with metastatic cells as polyclonal cell clusters. Clarifying the whole process of polyclonal metastasis will require investigating such early selection steps at the primary site.

In the present study, we showed that fibrotic niche generation was significantly suppressed in *Tgfbr2* knockout mouse liver, which was associated with reduced tumor growth. However, it has also been reported that TGF-β inhibitor treatment suppressed liver metastases of CRC cells through the activation of anti-tumor immunity[30]. Accordingly, we need to consider the potential contribution of the immune response to cancer cells together with the suppression of niche generation for reducing tumor growth following *Tgfr2* deletion.

In the present study, we also found that HSCs generate a fibrotic niche by proliferating outside and inside of vessel walls (Fig. 4c). Accordingly, tumor cells may have colonized and proliferated inside vessels where they were arrested in the model used in the present study (Figs. 4b and 7h). It is possible that co-transplanted Matrigel affected the process of the intra-vessel arrest of tumor cells; however, the results also suggest that malignant cancer cells can stimulate HSCs from inside vessels for the generation of a metastatic niche. Taken together, these results indicate that the regulation of fibrotic niche generation by targeting the pathway of metastatic cancer cell-secreted factors is an effective strategy against the development of polyclonal metastasis.

## Methods

**Organoid culture**. The intestinal tumor-derived organoids used in this study were established previously[8–10]. In brief, A, AK, AT, AP, and AKTP organoids were developed from intestinal tumors of $Apc^{\Delta716}$, $Apc^{\Delta716} Kras^{G12D}$, $Apc^{\Delta716} Tgfbr2^{-/-}$, $Apc^{\Delta716} Trp53^{R270H}$, and $Apc^{\Delta716} Kras^{G12D} Tgfbr2^{-/-} Trp53^{R270H}$ mice, respectively. These organoid cells were cultured in Growth Factor Reduced (GFR)-Matrigel (Corning) with Advanced DMEM/F-12 medium (Gibco) supplemented with 10 mM HEPES, 2 mM Glutamax, 1×B27, 1×N2 (Invitrogen), 100 ng/ml murine Noggin (Peprotech), and 1 μM N-acetylcysteine (Sigma) or cultured on dishes with Advanced DMEM/F-12 medium (Gibco) supplemented with 5 μM ALK inhibitor (A83-01), 5 μM GSK inhibitor (CHIR; Tocris Bioscience, UK), and 10 μM ROCK inhibitor (Y27632) (Wako, Osaka, Japan). A, AK, and AT cells were

cultured in Matrigel, while AP and AKTP cells were cultured either in Matrigel or on culture dishes.

For passaging of organoids, organoid cells were recovered from Matrigel with Cell Recovery Solution (Corning) and then mechanically dissociated by pipetting using 1-ml tips, split and passaged into fresh Matrigel. For enzymatic dissociation of organoid cells, cells were treated with 0.25% trypsin at 37 °C for 5 min, and dissociated cells were then filtered with a 35-μm mesh cell strainer (Falcon). Enzymatic dissociation was performed for AP, AKTP, and AKTP-DTR cells in all transplantation experiments except for Fig. 2 and Supplementary Fig. 2. A, AK, and AT cells were mechanically dissociated to avoid cell death by anoikis.

To label organoid cells and HSCs with fluorescent proteins, Venus and tdTomato cDNAs were subcloned to pPB-CAG-IP PiggyBac transposon expression vector (a kind gift from Hitoshi Niwa, Kumamoto University, Japan) and co-transfected with transposase expression vector to organoid cells using Lipofectamine (Thermo Fisher Scientific). Transfected clones were then selected by drug selection with 1 or 20 μg/ml of puromycin for organoid cells or HSCs, respectively. To construct DTR-expressing AKTP (AKTP-DTR) cells, cDNA of mutant heparan-binding EGF (HB-EGF I117L/L148V) (TransGenic Inc., Kobe, Japan) was subcloned to pPB-CAG-IB. DTR expression vector was co-transfected with a Venus expression vector and transposase expression vector to AKTP cells using Lipofectamine 3000 (Thermo Fisher Scientific), and transfected clones were selected using 1 μg/ml of puromycin and 10 μg/ml of blasticidin to establish a Venus-labeled AKTP-DTR line. Mycoplasma testing was performed using an indirect immunofluorescence test.

**Mouse experiments**. NOD/Shi-*scid Il2rg-/-* (NSG) mice were purchased (Charles River), and B6.129S6-*Tgfbr2*tm1Hlm/Nci (*Tgfbr2* flox) mice[31], 129S4-*Trp53*tm3Tyj/Nci (*Trp53* LSL R270H) mice[32], and B6;129-Gt(ROSA)26Sor tm1(cre/Esr1)Tyj/Nci (ROSA-CreER) mice[33] were obtained from NCI Mouse Repository (Frederick National Laboratory for Cancer Research). The mice were housed in specific-pathogen-free (SPF) conditions with 12-h light:dark cycle at 23 °C ± 2 °C temperature with relative humidity of 50 ± 20%, and given ad-libitum access to food and water for the duration of the study.

For the control metastasis experiments (Fig. 1), $2 \times 10^5$ AKTP-Venus and AKTP-tdTomato cells were mixed and injected into the NSG mouse spleen with 25 μl of Matrigel (Corning). Liver tissues were examined histologically at days 3, 7, and 14 or by CUBIC at 7 weeks after transplantation.

For the day-28 analysis, $1 \times 10^5$ AKTP-Venus and AKTP-tdTomato cells were mixed and injected ($n = 3$–4 mice for each experimental condition).

For the polyclonal metastasis experiments (Fig. 2), $1 \times 10^6$ organoid cells of two different genotype cells ($2 \times 10^6$ in total) were mixed and injected into the NSG mouse spleen with 25 μl of Matrigel. At 4 weeks after transplantation, liver metastases were examined histologically ($n = 3$–5 mice for each genotype combination).

For AKTP-DTR cell depletion experiments (Fig. 3), $1 \times 10^6$ AP or AT cells were mixed with $1 \times 10^5$ or $1 \times 10^6$ AKTP-DTR cells, respectively, and injected into the NSG mouse spleen with 25 μl of Matrigel. Mice were then intraperitoneally injected with 16.6 μg/kg of DT twice a week for 5 continuous weeks starting from 2 weeks after spleen transplantation. Liver tissues were examined by CUBIC imaging or histologically at 7 weeks after transplantation ($n = 4$–6 mice for each experimental combination).

For the chronological analysis at the early stage of colonization (Fig. 4), $1$–$5 \times 10^5$ or $1 \times 10^6$ of AKTP or AP/AT cells, respectively, were injected into the NSG mouse spleen with 25 μl of Matrigel. Liver tissues were examined histologically at days 1, 3, 7, and 14 after transplantation. As a control, only Matrigel (25 μl) was injected into the NSG mouse spleen, and liver tissues were examined histologically ($n = 3$ mice for each genotype at each time point).

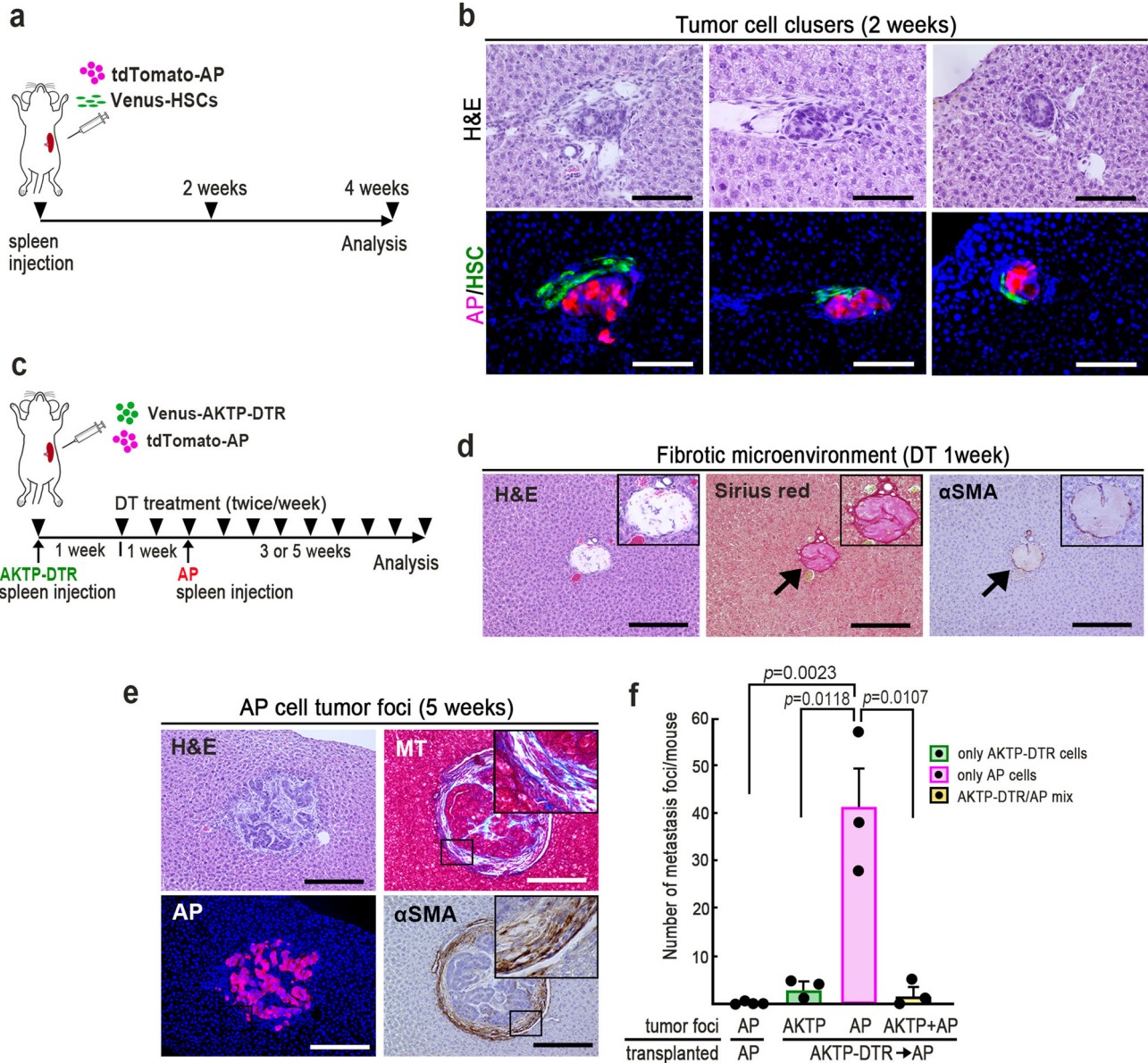

**Fig. 5 Fibrotic niche generation for non-metastatic cell colonization. a** A schematic illustration of the transplantation experiment with hepatic stellate cells (HSCs). tdTomato-labeled AP cells (red) and Venus-labeled HSCs (green) were mixed and transplanted to the spleen, and liver tissues were examined histologically at two and four weeks after transplantation. **b** Representative photographs of three independent tumor cell clusters in the liver at two weeks after transplantation. H&E (top) and fluorescent immunohistochemistry for AP cells (red) and HSCs (green) (serial sections, bottom). Bars, 100 μm. **c** A schematic illustration of the transplantation experiment for prior niche generation. Venus-labeled AKTP-DTR cells (green) were transplanted to the spleen, and mice were treated with diphtheria toxin (DT) twice a week starting from one week after transplantation. At one week after DT started, tdTomato-labeled AP cells (red) were transplanted to the spleen, and liver tissues were examined at three to five weeks later. **d** Representative photographs of H&E (left), Sirius red staining (center), and immunohistochemistry for αSMA (right) at one week after DT treatment. Insets show enlarged images. Arrows indicate deposited collagen fibers (center) and αSMA-positive cells (right). Bars, 100 μm. **e** Representative photographs of tumor lesions consisting of AP cells at five weeks after AP cell transplantation. H&E, Masson's trichrome staining (MT) and immunohistochemistry for tdTomato (AP) and αSMA. Insets indicate enlarged images of the boxed areas. Bars, 100 μm. **f** The numbers of liver metastasis foci consisting of only AKTP-DTR cells (green), only AP cells (magenta), and a mixture of AKTP-DTR and AP cells (yellow) at three to five weeks after AP cell transplantation are shown as a bar graph. The numbers of metastasis foci at four weeks after AP cell transplantation to control mice are also indicated (left). (*n* = 4 biologically independent animals for control; and *n* = 3 biologically independent DT-treated animals for each experimental condition.) The data in **f** are presented as the mean ± s.d. A two-sided unpaired *t*-test was used to calculate the significance of differences, and *p* values are provided. The photographs in **b**, **d**, and **e** are representative images from *n* = 3 biologically independent animals. Source data are provided as a Source Data File.

For serial transplantation experiments (Supplementary Fig. 3), $2.5 \times 10^5$ of Venus-labeled AKTP cells were injected into the spleen of NSG mice, followed by direct injection of $5 \times 10^5$ of tdTomato-labeled AKTP, AT, or AP cells into the portal vein at 2 weeks after spleen transplantation. Liver tissues were examined histologically at 5 weeks after spleen transplantation (*n* = 4–6 mice for each genotype combination).

For co-transplantation experiments using HSCs and AP cells (Fig. 5), $3 \times 10^5$ of Venus-labeled HSCs and AP cells were co-cultured overnight on ultra-low-attachment culture dishes (Corning) to generate chimeric aggregations and transplanted to the NSG mouse spleen. Liver tissues were examined histologically at 2 and 4 weeks after transplantation (*n* = 3 mice for each experiment).

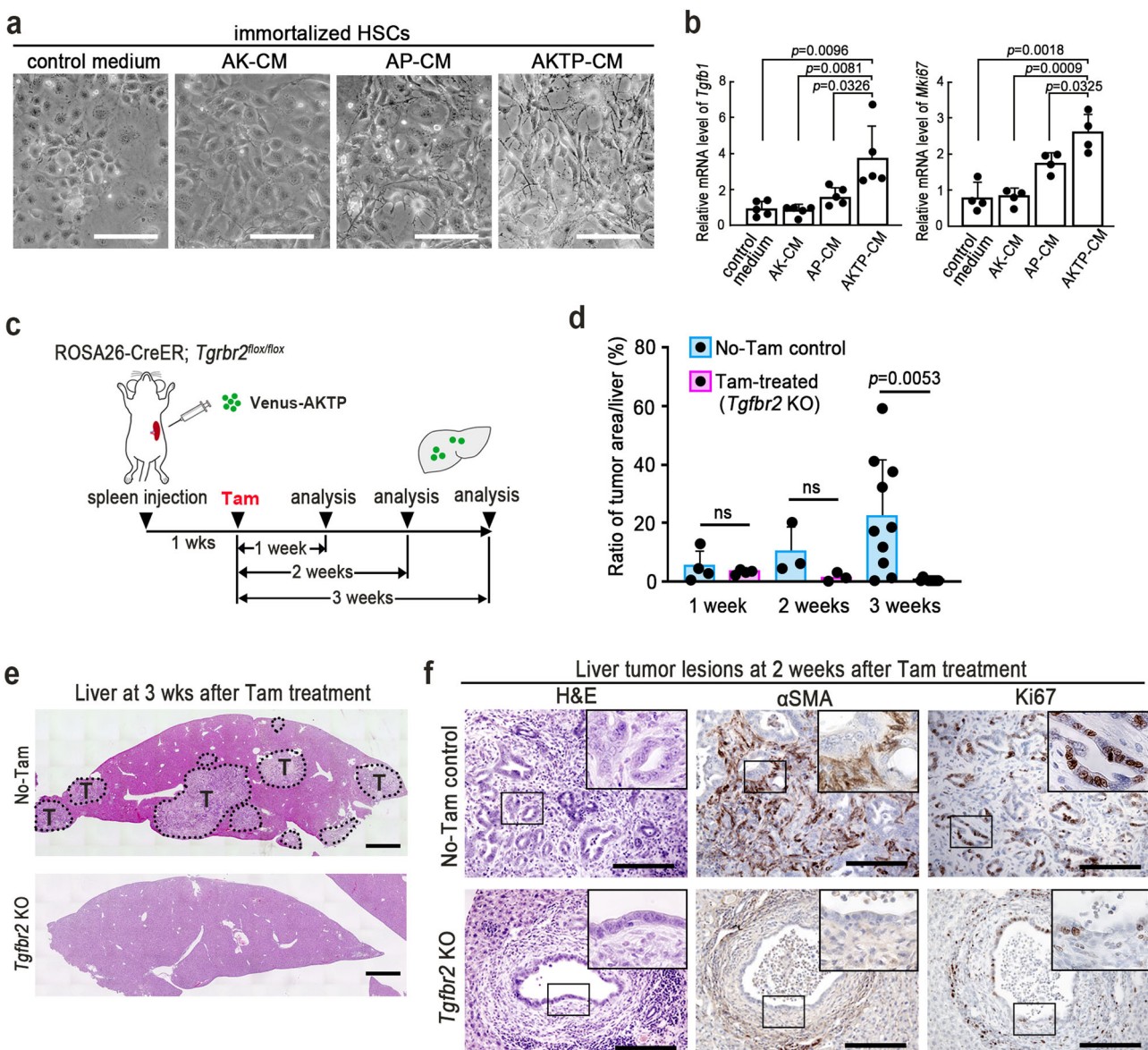

**Fig. 6 Fibrotic niche generation by hepatic stellate cell (HSC) activation. a** Representative photographs of HSCs cultured with control medium or conditioned medium (CM) of AK, AP, and AKTP cells are shown. Images are representative of $n = 5$ independent cultures. Bars, 100 μm. **b** Relative mRNA levels of *Tgfb1* (left) and *Mki67* (right) in HSCs cultured with control medium or AK-CM, AP-CM, or AKTP-CM are indicated as bar graphs ($n = 5$ and 4 biologically independent samples for *Tgfb1* and *Mki67* analyses, respectively). **c** A schematic illustration of the transplantation experiment. Venus-labeled AKTP cells (green) were transplanted to the spleen of ROSA26-CreER *Tgfbr2* flox mice. Mice were treated with tamoxifen (Tam) at one week after transplantation, and liver tissues were examined at one, two, and three weeks after Tam treatment. **d** The ratios of tumor areas in No-Tam control and Tam-treated *Tgfbr2*$^{-/-}$ (*Tgfbr2* KO) mouse livers measured on the histology sections are shown as a bar graph ($n = 4$ and 3 biologically independent animals analyzed at 1 and 2 weeks, respectively; and $n = 10$ and 8 biologically independent No-tam control and Tam-treated *Tgfbr2* KO animals analyzed, respectively, at 3 weeks). **e** Representative photographs of whole-liver images at two weeks after Tam treatment of No-Tam control (top) and Tam-treated *Tgfbr2* KO mice (bottom). The tumor areas are indicated by dotted lines. T tumor. Bars, 1 mm. **f** Representative photographs of liver tumor lesions at two weeks after Tam treatment. H&E (left), immunohistochemistry for αSMA (center) and Ki67 (right) of No-Tam control (top) and Tam-treated *Tgfbr2* KO mice (bottom). Insets indicate enlarged images of the boxed areas. Bars, 100 μm. The photographs in **e** and **f** are representative images from $n = 3$ biologically independent animals. The data in **b** and **d** are presented as the mean ± s.d. A two-sided unpaired *t*-test was used to calculate the significance of differences. *p* values are provided. ns not significant. Source data are provided as a Source Data File.

For the fibrotic niche analysis (Fig. 5), $1 \times 10^5$ of AKTP-DTR cells were injected into the NSG mouse spleen with 25 μl of Matrigel, and mice were intraperitoneally injected with 16.6 μg/kg of DT twice a week starting 1 week after spleen transplantation until the analysis time point. At 1 week after DT treatment started, fibrotic niche generation was examined histologically, and $1 \times 10^6$ AP cells were injected into the spleen with 25 μl of Matrigel, and liver tissues were examined histologically at 3–5 weeks after second transplantation ($n = 3$ mice for each experimental condition).

For the host TGF-β signaling analysis (Fig. 6), $1 \times 10^5$ AKTP-Venus cells with 25 μl of Matrigel were injected into the spleen of ROSA-CreER *Tgfbr2* flox compound mice. These mice were intraperitoneally treated with 4 mg/mouse of tamoxifen (Tam) at 1 week after transplantation to deplete the *Tgfbr2* gene. Liver tissues were then examined histologically at 1, 2, and 3 weeks after Tam treatment ($n = 3$ mice for examination at 1 and 2 weeks and 8–10 mice for examination at 3 weeks). The experimental protocols of all animal experiments were approved by the Committee on Animal Experimentation of Kanazawa University.

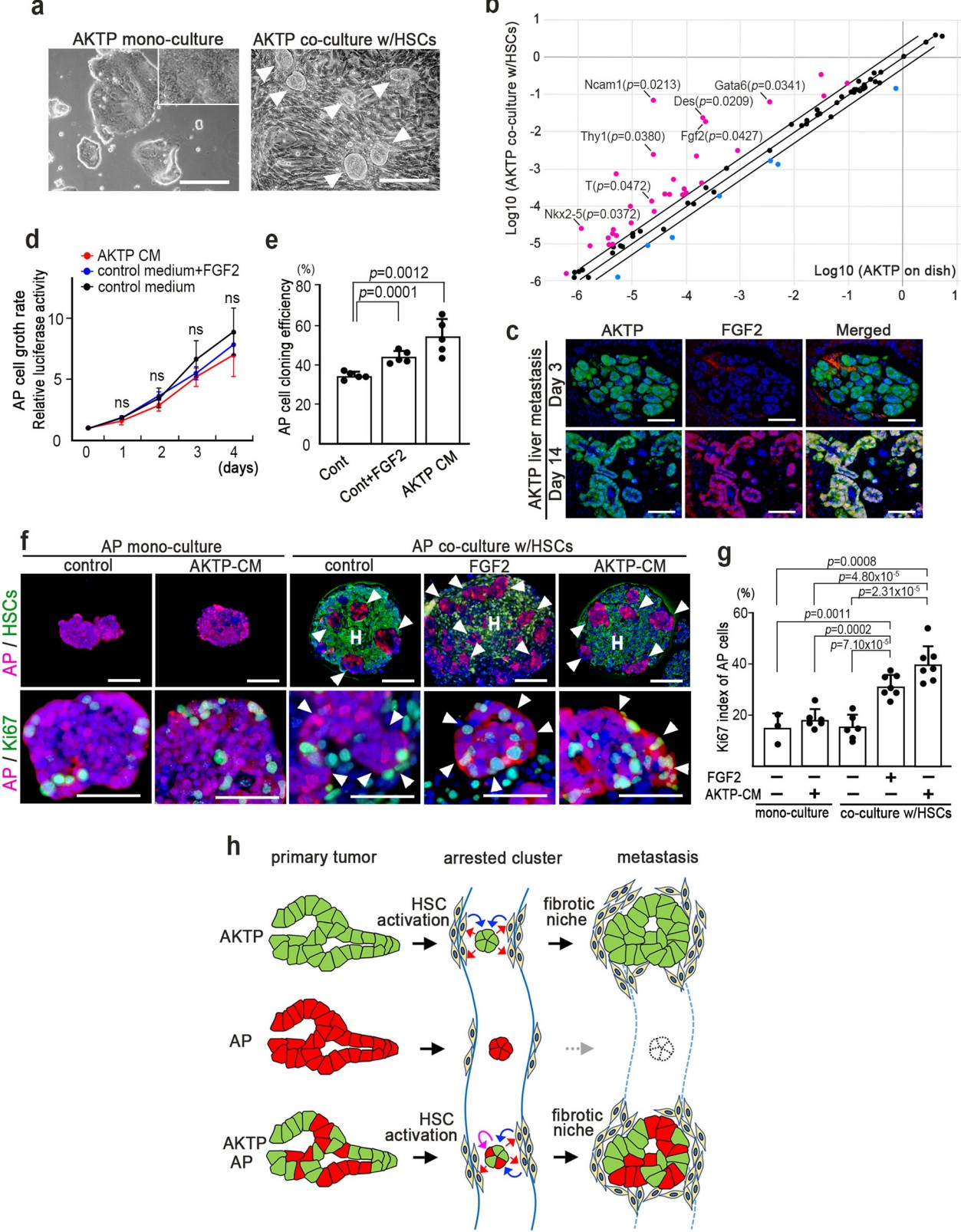

**Histology and immunohistochemistry**. Whole-liver lobules were cut at 4-mm intervals of thickness and fixed in 4% paraformaldehyde. All tissue specimens were then paraffin-embedded, and 4-μm-thick histology sections were prepared. For the histological analyses, the sections were stained with H&E. Antibodies against Ki67 (1:1000, Abcam; 1:50, BD Biosciences), GFP (1:500, MBL), RFP (1:1000, Rockland Immunochemicals), DsRed (1:200, Clonetech), αSMA (1:800, Sigma), CD31 (1:20, Clone SZ31; Dianova, Hamburg, Germany), and FGF2 (1:100, OriGene) were used

as primary antibodies for immunohistochemistry. Antibodies against GFP and RFP were used to detect Venus and tdTomato-labeled cells, respectively. Staining signals were visualized using a Vectastain Elite Kit (Vector Laboratories). A VECTOR Red Alkaline Phosphatase Substrate Kit (Vector Laboratories) was used for double-immunohistochemical staining to detect Ki67 and RFP. Alexa Fluor 594- or Alexa Fluor 488-conjugated antibodies (1:200, Molecular Probes) were used as secondary antibodies for fluorescent immunohistochemistry (FIHC). To detect collagen

**Fig. 7 Secreting factors from metastatic cells for polyclonal metastasis. a** Representative photographs of AKTP mono-culture and co-culture with hepatic stellate cells (HSCs). Images are representative of $n = 5$ independent cultures. The inset shows an enlarged image. Arrowheads indicate ES cell-like colonies. Bars, 100 μm. **b** Results of PCR array analysis (average Log10 of 3 independent samples). Magenta and blue, upregulated and downregulated genes, respectively, in AKTP cells co-cultured with HSCs. **c** Fluorescent immunohistochemistry for AKTP cells (green, left), FGF2 (red, center), and merged images with DAPI nuclear staining (right) of the AKTP cell-arrested liver vessels at days 3 (top) and 14 (bottom) after spleen transplantation. Bars, 100 μm. The photographs in **c** are representative images from $n = 3$ biologically independent animals. Bars, 100 μm. **d** Growth rates of AP cells stimulated with AKTP conditioned medium (CM) or FGF2 ($n = 3$ biologically independent samples for each culture condition). **e** Cloning efficiency of AP cells cultured with AKTP-CM or FGF2 ($n = 5$ plates of independent 96-well plate for each culture condition). **f** Representative photographs of fluorescent immunohistochemistry of AP mono-culture (left) and co-culture with HSCs (right) in the presence or absence of AKTP-CM or FGF2. tdTomato-labeled AP cells (red)/Venus-labeled HSCs (green) (top) and tdTomato-labeled AP cells (red)/Ki67 (green) (bottom) with DAPI nuclear staining. Bars, 100 μm (top) and 50 μm (bottom). Arrowheads, AP cells; H, HSCs in chimeric spheroids. **g** Ki67-labeling indices of AP cells in monoculture or co-culture with HSCs in the presence or absence of AKTP-CM or FGF2 (experiments in **f**, **g**: $n = 3$ and 6 biologically independent samples for AP mono-culture in the absence and presence of AKTP-CM, respectively; $n = 6$ and 7 biologically independent samples for control and FGF2- or AKTP-CM-treated AP cell co-culture, respectively). The data in **d**, **e**, and **g** are presented as the mean ± s.d. The data in **d** were analyzed by one-way ANOVA test, ns not significant. The data in **e** and **g** were analyzed by a two-sided unpaired $t$-test, and $p$ values are provided. **h** Schematic illustration of possible polyclonal metastasis mechanism. Source data are provided as a Source Data File.

---

fibers, sections were stained with Masson's trichome using Trichrome Stain Kit (Sigma) and Sirius red using Picrosirius Red Stain Kit (Polysciences).

The ratio of Venus-labeled AKTP cells and tdTomato-labeled AKTP, AT, or AP cells in the fluorescent immunohistochemistry sections ($n = 3$ for each genotype combination) was scored by counting the number of cells in each tumor lesion ($n = 72$, 117, and 186 for co-transplantation of Venus-AKTP/tdTomato-AKTP, Venus-AKTP/tdTomato-AT, and Venus-AKTP/tdTomato-AP cells, respectively) using an All-In-One microscope (Keyence, Osaka, Japan). The tumor areas in the H&E histology sections were measured using ImageJ application (NIH). The mean Ki67-labeling indices of tumor cells or αSMA-positive cells were calculated by counting the number of Ki67-positive cells or αSMA-positive cells in 5 independent microscopic fields of metastatic liver tumors ($n = 3$ mice for each genotype).

**Whole-liver tissue imaging by CUBIC**. CUBIC clearing solutions were prepared as follows[19]. CUBIC-L solution for decolorization and delipidation was prepared as a mixture of 10% (w/w) polyethylene glycol mono-$p$-isooctylphenyl ether/Triton X-100 (Nacalai Tesque, Kyoto, Japan) and 10% (w/w) $N$-buthyldiethanolamine (Tokyo Chemical Industry, Tokyo, Japan). CUBIC-R solution for RI matching was prepared as a mixture of 45% (w/w) 2,3-dimethyl-1-phenyl-5-pyrazolone/anti-pyrine and 30% (w/w) nicotinamide (Tokyo Chemical Industry). Liver tissues were fixed by perfusion of 4% paraformaldehyde (PFA) via the left ventricle of the heart, and tissues were post-fixed in 4% PFA at 4 °C. After washing in phosphate-buffered saline (PBS), the fixed tissues were immersed in 50% CUBIC-L at 37 °C for more than 6 h followed by 100% CUBIC-R for 3–5 days. After washing in PBS, tissues were immersed in 50% CUBIC-R for 6 h followed by 100% CUBIC-R for more than 1 day. Whole-liver tissue images were acquired with a custom-built light-sheet fluorescence microscope (LSFM; developed by Olympus, Tokyo, Japan). Laser bandwidths of 488 nm and 532 nm/590 nm were used to detect Venus and tdTomato, respectively. Three-dimensionally rendered images were visualized, captured, and analyzed with the Imaris software program (ver 8.4, Bitplane AG, Zurich, Switzerland) and Free Imaris Viewer (ver 9.5, Bitplane AG).

**Cell culture experiments**. Hepatic stellate cells (HSCs) were isolated from homozygous *Trp53* LSL R270H mouse liver as follows[34]. Liver tissues of mice were treated with 100 units/ml collagenase IV (Thermo Fisher) by perfusion through the portal vein, excised and incubated in collagenase IV solution for another 30 min to isolate HSCs. Collected cells were cultured in high-glucose DMEM containing 10% FBS. HSCs were identified by their myofibroblast-like characteristic morphology, and the vimentin and αSMA expression was confirmed by immunohistochemistry at an early passage. To prepare conditioned medium (CM) of the respective genotype tumor cells, $2 \times 10^6$ cells were cultured in 8 ml of control medium (i.e., Advanced DMEM/F12 with 10% FBS), 5 μM GSK inhibitor (CHIR), and 10 μM ROCK inhibitor (Y27632) for 48 h. CM was then collected, and cells were cultured with fresh medium for another 48 h followed by the further collection of CM. The collected CM was pooled, centrifuged at 420 x $g$ for 5 min and filtered through a Millex 0.22-μm filter (Millipore). CM was added to HSC culture media with control media at a ratio of 2:1 and cultured for 72 h before collection of RNA and protein.

For AKTP and HSC co-culture experiments, AKTP cells and HSCs were mixed at the ratio of 4:1, and cultured on dishes in DMEM with 10% FBS for 96 h. AKTP cell colonies that formed on the HSC layer were picked up using fine forceps and used for the expression analysis. Monoculture of AKTP cells was used as a control.

For AP and HSC co-culture experiments, AP cells and HSCs were mixed at a ratio of 1:1, cultured on an Ultra-Low Attachment Multiple Well Plate (Corning) in control medium with or without AKTP-CM, and cultured for 4 days to form

spheroids before being embedded in iPGell (GenoStaff). Sections of spheroids embedded in iPGell were examined by immunohistochemistry for GFP, DsRed, and Ki67.

For the cell growth assay, $1 \times 10^3$ cells were seeded on 96-well plates in the presence of 100 ng/ml DT (Sigma), AKTP-CM or 10 ng/ml FGF2 (Wako), and the cell viability was examined at 0, 0.5, 1, 2, and 3 days using CellTiter-GLO (Promega) according to the manufacturer's protocol.

For AP cell cloning efficiency, single cells were seeded in each well of 96-well plates, and the numbers of wells with colonies were scored at 3 weeks to calculate the ratio of cloning efficiency ($n = 5$ independent 96-well plates for each culture condition).

**Reverse transcription polymerase chain reaction (RT-PCR)**. HSCs were cultured with CM or control medium. AKTP cells were co-cultured with HSCs using cell culture inserts (Falcon) to avoid cross contamination of cells. Total RNAs were extracted from these cells using the ISOGEN reagent (Nippon Gene, Toyama, Japan) or RNeasy Plus Micro Kit (Qiagen), reverse-transcribed using a PrimeScript RT reagent Kit (Takara Bio, Otsu, Japan), and amplified using ExTaqII SYBR Premix (Takara) on a Mx3000P real-time thermocycler (Agilent Technologies). Relative mRNA levels of *Tgfb1*, *Mki67*, and *Fgf2* to the mean *Gapdh* were calculated. The primer sequences are provided in Supplementary Table 1.

**PCR array analysis**. Total RNA was isolated from co-cultured AKTP cells with HSCs and control mono-culture AKTP cells using an RNeasy Plus Micro Kit (Qiagen GmbH, Hilden, Germany). The cDNA was synthesized and pre-amplified using an RT² PreAMP cDNA Synthesis Kit (Qiagen) before the microarray analysis. An expression analysis was performed using the RT² Profiler PCR Array Mouse Embryonic Stem Cells (PAMM-081YA, Qiagen). The statistical significance of the expression data was calculated using the GeneGlobe Data Analysis Center, a web-based online tool for the RT-PCR results (https://geneglobe.qiagen.com/us/analyze/).

**Statistical analyses**. The data were analyzed using two-sided unpaired $t$-tests otherwise mentioned and presented as the means ± standard deviation (s.d.). Statistical analysis for Fig. 7d and Supplementary Fig. 4b was performed using a one-way ANOVA test. A value of $p < 0.05$ was considered to be statistically significant. Excel (16.23, Microsoft) and Graphpad Prism7 (GraphPad) were used for statistical analyses. All data were reproduced with at least two independent experiments and at least three biological replicates.

**Reporting summary**. Further information on research design is available in the Nature Research Reporting Summary linked to this article.

## Data availability

The source data underlying Figs. 1b, c, 2c, e, 3f, 4e, g, 5f, 6b, d, 7b, d, e, g and Supplementary Figs. 2b, 4b, 5, 8 are provided as a Source Data file. All other data supporting the findings of this study are available within the article and its supplementary data and from the corresponding author upon reasonable request. The GeneGlobe Data Analysis Center was used for statistical analysis of PCR Array data (https://geneglobe.qiagen.com/us/analyze/). Source data are provided with this paper.

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

## Acknowledgements

We thank Manami Watanabe, Ayako Tsuda, and Yoshie Jomen for their technical assistance. This work was supported by AMED (19ck0106259h0003, 20ck0106541h0001) (M.O.) and AMED (19dm0207049, 19am0401011) (H.R.U.) from the Japan Agency for Medical Research and Development, Japan; and Grants-in-Aid for scientific Research (A) (18H04030) (M.O.), (B) (19H03498) (H.O.), (S) (25221004) (H.R.U), Grant-in-Aid for Scientific Research on Integrated Analysis and Regulation of Cellular Diversity (17H06326) (K.Miyazono), and Grant-in-Aid for Early-Career Scientists (19K16604) (K.T.) from the Ministry of Education, Culture, Sports, Science and Technology of Japan.

## Author contributions

S.Y.K., H.O., and M.O. conceived the study. S.Y.K., H.O., and M.O. designed the experiments. S.Y.K. and H.O. performed the organoid transplantation experiments. K.T., H.R.U., and K. Miyazono performed the imaging of cleared tissues. M.N. generated the organoids. K. Murakami constructed the fluorescent-labeled organoids.

## Competing interests

The authors declare no competing interests.
