## [Peer Review File · Nature Communications]

REVIEWER COMMENTS

Reviewer #1 (Expertise: CRC, liver metastasis, Remarks to the Author):

Kok et al generated mouse intestinal tumor organoids from cells carrying colon cancer driver mutations including single *Apc*D716 (A), *Kras*G12D (K), *Tgfbr2*^{-/-} (T) and *Trp53*R270H (P) mutations and combinations thereof. Here they used organoid-derived cells to investigate mechanisms underpinning polyclonal metastases and the role of a dominant metastatic clonal population (combined AKTP mutations) in enabling liver metastasis of non-metastatic clones (AP and AT mutations only). They show that when non-metastatic AP or AT cells were transplanted into the spleen (in Matrigel) together with highly metastatic AKTP cells, they formed chimeric metastatic foci in the liver. Using AKTP cells expressing-diphtheria toxin receptor (AKTP-DTR), they found that once liver metastases were established, non metastatic AP cells continued to proliferate, even after depletion of AKTP cells by diphtheria toxin injections. The authors show that AKTP cells-activated hepatic stellate cells produced a “fibrotic niche” and propose that these niches were essential for expansion of the polyclonal metastases that include proliferating AP and AT clones.

Overall, this is an interesting and generally well executed and controlled study that provides insight into early events in liver metastasis, particularly the factors that regulate the establishment of polyclonal vs. monoclonal metastases. However, the results raise a few important issues that need to be addressed.

Major Issues to be addressed:

1. Fig. 1. The density of metastatic colonies in the livers shown in this figure is very high. Under these experimental conditions and based on fluorescent imaging alone, truly chimeric metastases are not easily distinguishable from those resulting from merged adjacent monoclonal metastases. This can be resolved by analyzing earlier time points or injecting less cells.
2. Fig. 3. To support their conclusion that AKTP cells initiate fibrotic niche formation and in this way, promote seeding and outgrowth of AP and AT cells, a more relevant experimental design is to allow DR-AKTP cells to seed and expand and then use diphtheria toxin to eliminate the cells before injecting AP or AT cells. This would provide conclusive evidence that the microenvironment generated by AKTP cells (i.e. HSC-derived ECM) can support metastatic expansion of the less invasive cells, even in the absence of the highly metastatic cells.
4. Fig. 3c&e. The non-DT controls for AP and AT colonies are different with a markedly lower baseline for AT cells. For a comparison of AT and AP growth after DT treatment, the results should be expressed as fold change relative to non-DR treated controls.

5. Figures 1-3. These figures lack statistical data. What is the “n”? Does it refer to the number of livers analyzed in a single experiment or the number of independent experiments performed? What was the variability among different livers analyzed?

6. Fig. 4 - The areas surrounding metastatic colonies are identified as “fibrotic niches” based on the accumulation of α SMA+ cells and HE staining. To identify these areas as fibrotic, a specific stain such as Sirius Red or ECM-specific antibodies should be used.

7. Fig. 4C. The authors show that “...in liver vessels without tumor cells ... α SMA -positive hepatic stellate cells (HSCs) lined outside the endothelial cell layer”. α SMA is a marker of activated (as opposed to quiescent) HSC. In the absence of tumor cells, what is activating these cells and why is ECM deposition absent? This should be addressed.

Also, the authors state that “We confirmed that Matrigel transplantation did not cause such fibrotic responses in the liver (data not shown)” This is an important control because Matrigel contains both extracellular matrix proteins and growth factors such as TGF β that could activate HSC. These data should be shown.

Fig. 5. The results shown raise the possibility that in addition to initiation of a fibrotic response, AKTP-cells promote metastatic expansion of AP and AT cells by producing growth factors such as FGF2 that induce the growth of these cells (or migration/invasion) in a paracrine fashion. Is FGF2, or are other growth factors produced by AKTP cells promoting the growth of the non-metastatic cells in the liver? Does FGF2 promote neovascularization? Neutralizing antibodies, chemical inhibitors or gene silencing can be used to address these questions.

Minor point

In the Discussion, the authors state that in this model tumor cells “were disseminated and arrested by disruption of the vessel structure rather than by extravasation to the liver parenchyma” and that “mechanical arrest rather than extravasation is the primary metastatic step in the liver metastasis (reference 27)”. This is an inaccurate citation of this publication, where mechanical arrest is described as leading to extravasation. Most likely the authors’ finding of intravascular expansion and vascular disruption are related to the mode of inoculation where cells were administered in Matrigel and disseminated as clusters rather than as single cells. The potential role of Matrigel in the mode of dissemination in the present model should be discussed.

Reviewer #2 (Expertise: Liver fibrosis, Remarks to the Author):

Kok et al investigate mechanisms by which tumor cells with different driver mutations achieve polyclonal metastasis and mechanisms that underlie the permissive effect of more aggressive tumors that allows less aggressive tumors to co-metastasize. The authors postulate that this is mediated by the activation of fibrogenic cells in the liver as demonstrated by the inability of less aggressive cells to induce this in vitro or in vivo. However, the manuscript contains significant weaknesses:

1. The first part of the manuscript is interesting and mechanistically supported, showing that non-metastatic cells can become metastatic when coinjected with a metastatic cell line, with various experimental approaches such as depletion of one population and different injection timing and routes to interrogate mechanisms. The second part on the underlying mechanisms is weak and lacks functional experiments. Moreover, the concept of polyclonal metastasis in the first part have already been established in other settings as mentioned by the authors in the introduction (Ref 4-7). Hence, the mechanistic aspect in part 2 is important for the novelty and relevance of this paper.
2. The authors postulate that HSCs play an important role in the permissive effect that metastatic cells exert on the growth of non-metastatic cells, but effectively lack evidence for this. The authors show accumulation of aSMA+ cells around the lesions, but do not show functional contributions of these cells.
3. The data in Fig.5A suggest in fact that HSCs are not sufficient to promote metastatic growth of AP tumor cells. Hence, it is likely that HSCs represent an important component of the niche - but that they are only a marker of advanced lesions or that they are necessary but not sufficient for metastatic growth.
4. The co-culture experiments in Fig.5C-D show effects of tumor cell conditioned media on HSC with the strongest effect of AKTP conditioned media, but do not show that this promotes the metastatic growth of tumors. Likewise, the data in Fig.5E-F show interesting phenotypic and gene expression changes upon co-culture, but do not reveal any functions of these in regards to the metastatic process.

Point-by-point explanation to the comments by Editor and Reviewers

Our point-by-point responses to the comments made by the Reviewers are included below. The comments are quoted “*verbatim*”, followed by our **responses**. Changes in the manuscript are **highlighted** in the revised text.

Reviewer 1: (Expertise: CRC, liver metastasis, Remarks to the Author):

[...] Overall, this is an interesting and generally well executed and controlled study that provides insight into early events in liver metastasis, particularly the factors that regulate the establishment of polyclonal vs. monoclonal metastases. However, the results raise a few important issues that need to be addressed.

Response: We thank Reviewer 1 for these positive and constructive comments on our manuscript and have responded to all of their comments as follows:

Major Issues to be addressed:

1. *Fig. 1. The density of metastatic colonies in the livers shown in this figure is very high. Under these experimental conditions and based on fluorescent imaging alone, truly chimeric metastases are not easily distinguishable from those resulting from merged adjacent monoclonal metastases. This can be resolved by analyzing earlier time points or injecting less cells.*

Response: As suggested, we reduced the number of transplanted Venus-labeled AKTP and AKTP-tdTomato cells to 1×10^5 and examined the metastatic foci at 4 weeks (28 days) after transplantation by fluorescence immunohistochemistry (FIHC). Consistent with the previous results at days 7 and 14, we found that multiple tumor foci had developed in the liver at day 28, consisting of a mixture of Venus- and tdTomato-labeled cells or either Venus- or tdTomato-labeled cells at similar ratios. We have now added the FIHC findings at day 28 in Fig. 1d and in the main text (page 6). To support the FIHC data, we included a bar graph indicating the ratios of Venus only, tdTomato only, and Venus/tdTomato mix tumor cell clusters at each time point in Fig. 1c. In addition, we have added a statement concerning CUBIC imaging to reflect the Reviewer’s comment as follows: “Although the chimeric metastasis were difficult to distinguish from merged adjacent monoclonal metastasis on CUBIC images, we confirmed the mixture of Venus- and tdTomato-labeled tumor cells in the metastatic lesions by immunohistochemistry (Fig. 1f)” (page 6).

2. *Fig. 3. To support their conclusion that AKTP cells initiate fibrotic niche formation and in this way, promote seeding and outgrowth of AP and AT cells, a more relevant experimental design is to allow DR-AKTP cells to seed and expand and then use diphtheria toxin to eliminate the cells before injecting AP or AT cells. This would provide conclusive evidence that the microenvironment generated by AKTP cells (i.e. HSC-derived ECM) can support metastatic expansion of the less invasive cells, even in the absence of the highly metastatic cells.*

Response: We would like to thank Reviewer 1 for giving us the idea to perform valuable transplantation experiments to provide conclusive evidence. We transplanted AKTP-DTR cells to the spleen, and at one week after transplantation, mice were treated with DT for a further week to eliminate AKTP-DTR cells. As suggested by Reviewer 1, we confirmed the remaining fibrotic niche in the liver vessels by Sirius red staining and α SMA IHC. As a secondary transplantation, AP cells were transplanted to the spleen of DT-treated mice, and mouse livers were examined at three to five weeks after transplantation. Notably, we found that AP cells survived and formed a significant number of colonies without AKTP cells. Such tumor lesions were not detected when AP cells were transplanted to control mice without the prior generation of fibrotic niche. These results indicate the role of a fibrotic niche generated by AKTP cells for the survival and colonization of non-metastatic AP cells. We have now added these results as Fig. 5c-f and in the main text (page 13-14). The Methods (page 24) and Figure legends (page 43-44) have been updated accordingly. We have also added the following to the Abstract: "...such fibrotic microenvironment promotes the colonization of AP cells" (page 2).

4. *Fig. 3c&e. The non-DT controls for AP and AT colonies are different with a markedly lower baseline for AT cells. For a comparison of AT and AP growth after DT treatment, the results should be expressed as fold change relative to non-DR treated controls.*

Response: As suggested by Reviewer 1, in the revised manuscript, we compared the size of AP and AT cell tumors of DT-treated mice with those of non-DT control mice and examined the size distribution changes. Through this analysis, we confirmed that the tumor size of AP cells was significantly larger in the DT-treated mice than in the

non-DT control mice, indicating that AKTP cells are not required for the survival and proliferation of AP cells and that AKTP cells may negatively regulate the proliferation of AP cells in chimeric tumors. In contrast, the size distribution of AT cells was not markedly changed, indicating that AKTP cells are not required for the AT cell survival. We have replaced Fig. 3f with new bar graphs indicating the size distribution changes in AP and AT cells and explained the results in the main text (page 10). Figure legends have been updated accordingly (page 41).

5. Figures 1-3. These figures lack statistical data. What is the “n”? Does it refer to the number of livers analyzed in a single experiment or the number of independent experiments performed? What was the variability among different livers analyzed?

Response: As indicated in the *Mouse experiments* subsection of the Methods, “n” indicates the number of mice used in each experiment. To avoid confusion, we have explained as “n=3-4 mice for each experimental condition”, “n=3-5 mice for each genotype combination”, and “n=4-6 mice for each experimental combination” in the Methods section (page 22-23). We also added information on the number of animals in the Figure legends where required.

To avoid a possible variation in metastatic tumor development in different liver lobules of the same mouse, we cut all lobules at 4-mm intervals of thickness before fixation and analyzed the histology of all specimens for all mice. We have now mentioned this in the *Histology and Immunohistochemistry* subsection of the Methods (page 25).

In addition, we performed statistical analyses and provided exact *p* values in Fig. 2c, 3f, 4e, 4g, 5f, 6b, 6d, 7c, 7d, 7f, and Supplementary Fig. 2b, 4b, and 5.

6. Fig. 4 - The areas surrounding metastatic colonies are identified as “fibrotic niches” based on the accumulation of aSMA+ cells and HE staining. To identify these areas as fibrotic, a specific stain such as Sirius Red or ECM-specific antibodies should be used.

Response: As suggested, we performed Masson’s trichrome staining and Sirius red staining to detect collagen fibers in the niche of AKTP cell-arrested liver vessels at 14 days after transplantation. We confirmed the fibrotic niche generation by positive staining of these methods. We have now added representative photographs of

Masson's trichrome and Sirius red staining as Fig. 4c and in the main text (page 11). The methods (page 25) and Figure legends (page 42) have been updated accordingly.

7. Fig. 4C. The authors show that "...in liver vessels without tumor cells ... α SMA-positive hepatic stellate cells (HSCs) lined outside the endothelial cell layer". α SMA is a marker of activated (as opposed to quiescent) HSC. In the absence of tumor cells, what is activating these cells and why is ECM deposition absent? This should be addressed.

Response: As the reviewer mentioned, α SMA is a marker of activated hepatic stellate cells (HSCs). However, we detected an α SMA-positive cell layer (thin layer) outside of the endothelial cells of the non-tumor region of AKTP tumor cell-transplanted mouse livers, as shown in the previous Fig. 4c. It is possible that HSCs were partially activated in the normal region by the effect of factors secreted by AKTP cells. However, we were unable to determine the cell-type of α SMA-positive cells by a histological analysis in this study. Therefore, we removed the results of fluorescent IHC for α SMA in the non-tumor region in the revised manuscript.

However, we showed that α SMA-positive cells were proliferating inside and outside of the endothelial cell layer at AKTP cell-arrested vessels (Fig. 4d), suggesting that the fibroblast-like cells in metastatic lesions originated from HSCs. We have now added the following to the main text: "Notably, α -smooth muscle actin (α SMA)-positive cells were found inside and outside of the AKTP cell-arrested vessel wall (Fig. 4d, left) and were proliferating (Fig. 4d, e). Accordingly, it is possible that the proliferating fibroblast-like cells in the metastatic niche originated from HSCs." (page 11).

8. Also, the authors state that "We confirmed that Matrigel transplantation did not cause such fibrotic responses in the liver (data not shown)" This is an important control because Matrigel contains both extracellular matrix proteins and growth factors such as TGF β that could activate HSC. These data should be shown.

Response: We have now added the results of a chronological histological analysis (H&E) of the liver vessels of mice transplanted with only Matrigel at day 1, 3, 7, and 14 after transplantation. We confirmed that host reactions, such as fibroblast-like cell

proliferation, were not detected in vessels where Matrigel had been arrested inside, excluding the possibility of Matrigel-induced fibrotic niche generation. We have now added these results as Fig. 4b (bottom) and explained them in the main text (page 11).

9. Fig. 5. The results shown raise the possibility that in addition to initiation of a fibrotic response, AKTP-cells promote metastatic expansion of AP and AT cells by producing growth factors such as FGF2 that induce the growth of these cells (or migration/invasion) in a paracrine fashion. Is FGF2, or are other growth factors produced by AKTP cells promoting the growth of the non-metastatic cells in the liver? Does FGF2 promote neovascularization? Neutralizing antibodies, chemical inhibitors or gene silencing can be used to address these questions.

Response: As suggested by Reviewer 1, AKTP cell-secreted factors, including growth factors, may directly promote the AP cell survival and colonization. In addition, we showed in the revision that the AKTP cell-generated niche plays a role in AP cell colonization (Fig. 5 of this revision). We therefore additionally examined both possibilities, namely the direct effect of AKTP-secreted factors and indirect effect of AKTP-secreted factors via HSC activation on the survival and proliferation of AP cells, by the following experiments:

1. AP cells were cultured with AKTP-conditioned medium (CM) or control medium+FGF2, one of the upregulated genes in AKTP cells by co-culture with HSCs, and examined the proliferation and cloning efficiency. Although the proliferation rate of AP cells (examined by the luciferase activity) was not changed by AKTP-CM or FGF2, the cloning efficiency of AP cells (examined by limiting dilution in a 96-well plate) was significantly increased by both AKTP-CM and FGF2. These results indicate that AKTP cell-derived factors promote the clonal expansion of AP cells by direct mechanism.
2. Next, AP cells were mono-cultured or co-cultured with HSCs by forming a chimeric spheroid (Fig. 7e) in the presence or absence of AKTP-CM or FGF2, and the Ki67 labeling index was examined. The proliferation of AP cells was not increased by either mono-culture with AKTP-CM or co-culture without AKTP-CM

(control medium). Importantly, however, AP cell proliferation increased significantly when co-cultured with HSCs in the presence of AKTP-CM or FGF2 (Fig. 7e, f), although AKTP-CM and FGF2 failed to increase AP cell proliferation without HSCs in culture. These results indicate that AKTP cell-derived factors including FGF2 indirectly promote AP cell proliferation through the activation of HSCs, which is consistent with the new *in vivo* results shown in Fig. 5c-f.

These novel results indicate that metastatic cells promote polyclonal metastasis both by direct effect and indirect mechanisms through HSC activation. Particularly, FGF2 plays an important role in clonal expansion of AP cells by direct mechanism and in proliferation of AP cells through activation of HSCs. We have now added these results as Fig. 7c-f and the main text (page 15-16). Furthermore, we have added text describing the concept of a polyclonal metastasis by a driver clone as indirect mechanism through HSC activation and direct mechanism by secreting factors to the Discussion section (page 17). The Methods (page 28, 29) and Figure legends (page 46-47) have been updated accordingly.

Although Reviewer 1 asked about the role of FGF2 in angiogenesis, we did not find any marked increase in angiogenesis at the early stage of metastasis, so we could not assess this point. We hope that the Reviewer understands this issue.

Minor point

In the Discussion, the authors state that in this model tumor cells “were disseminated and arrested by disruption of the vessel structure rather than by extravasation to the liver parenchyma” and that “mechanical arrest rather than extravasation is the primary metastatic step in the liver metastasis (reference 27)”. This is an inaccurate citation of this publication, where mechanical arrest is described as leading to extravasation. Most likely the authors’ finding of intravascular expansion and vascular disruption are related to the mode of inoculation where cells were administered in Matrigel and disseminated as clusters rather than as single cells. The potential role of Matrigel in the mode of dissemination in the present model should be discussed.

Response: We apologize for including the wrong citation of the reference concerning colonization in the sinusoid without extravasation. We agree that stacked Matrigel inside vessels may contribute to vascular disruption and the expansion of arrested cancer cells in the model used in the present study. However, the present results also suggest that AKTP cells can stimulate HSCs from inside vessels for the generation of a metastatic niche. We removed the original sentence with the wrong citation and have instead discussed the possible role of Matrigel as well as the possible mechanism underlying niche generation by arrested AKTP cells in the Discussion, as follows: “Accordingly, tumor cells may have colonized and proliferated inside vessels where they were arrested in the model used in the present study (Fig. 4b and 7g). It is possible that co-transplanted Matrigel affected the process of the intra-vessel arrest of tumor cells; however, the results also suggest that malignant cancer cells can stimulate HSCs from inside vessels for the generation of a metastatic niche.” (page 18).

Reviewer #2 (Expertise: Liver fibrosis, Remarks to the Author):

Kok et al investigate mechanisms by which tumor cells with different driver mutations achieve polyclonal metastasis and mechanisms that underlie the permissive effect of more aggressive tumors that allows less aggressive tumors to co-metastasize. The authors postulate that this is mediated by the activation of fibrogenic cells in the liver as demonstrated by the inability of less aggressive cells to induce this in vitro or in vivo. However, the manuscript contains significant weaknesses:

Response: We thank Reviewer 2 for the constructive comments on our manuscript. We have responded to these comments as follows:

1. The first part of the manuscript is interesting and mechanistically supported, showing that non-metastatic cells can become metastatic when coinjected with a metastatic cell line, with various experimental approaches such as depletion of one population and different injection timing and routes to interrogate mechanisms. The second part on the underlying mechanisms is weak and lacks functional experiments. Moreover, the concept of polyclonal metastasis in the first part have already been established in other settings as mentioned by the authors in the introduction (Ref 4-7). Hence, the mechanistic aspect in part 2 is important for the novelty and relevance of this paper.

Response: As the reviewer pointed out, the concept of polyclonal metastasis has already been proposed by several reports based on animal experiments (Ref. 14-16). However, how genetic alterations in cancer cells are involved in polyclonal metastasis has not yet been addressed.

In the present study, we examined the mechanism of polyclonal metastasis using genetically defined and phenotypically characterized tumor organoid cells and found a correlation between genetic alteration patterns and polyclonal metastasis phenotypes (i.e. AKTP quadruple-mutant cells but not AK, AT, or AP double-mutant cells can drive polyclonal metastasis, and AP and AT invasive cells but not A or AK non-invasive cells can metastasize by a polyclonal mechanism with a driver clone, suggesting a requirement of invasiveness for polyclonal metastasis). We also showed that non-metastatic AP cells can form large tumors without a driver clone once a metastatic lesion is established. This point is unexpected and particularly important for determining the ideal clinical strategy as described in Discussion (page 17). We

believe that these results will also help expand our knowledge concerning the metastasis mechanism.

However, as Reviewer 2 indicated, the mechanistic insight was weak in the original manuscript. Therefore, we additionally performed *in vitro* and *in vivo* experiments to respond to subsequent comments.

2. The authors postulate that HSCs play an important role in the permissive effect that metastatic cells exert on the growth of non-metastatic cells, but effectively lack evidence for this. The authors show accumulation of aSMA+ cells around the lesions, but do not show functional contributions of these cells.

Response: To examine the functional contribution of activated HSCs for polyclonal metastasis, we additionally performed the following three experiments:

1. Non-metastatic AP cells were mono-cultured or co-cultured with HSCs by forming a chimeric spheroid in the presence or absence of AKTP-conditioned media (CM) or one of AKTP-secreting factors, FGF2 (Fig. 7e), and the Ki67 labeling index was examined (same experiment as mentioned in our response to Reviewer 1's comment 9). The proliferation of AP cells was not increased by either mono-culture with AKTP-CM or co-culture without AKTP-CM (control medium). Importantly, however, it increased significantly when AP cells were co-cultured with HSCs in the presence of AKTP-CM as well as FGF2 (Fig. 7f), although AKTP-CM and FGF2 failed to increase AP cell proliferation without HSCs in culture (Fig. 7c). These results indicate that AKTP cell-derived factors indirectly promote AP cell proliferation through the activation of HSCs. We have now added these results as Fig. 7c, e, f and in the main text (page 15-16). The Methods (page 28) and Figure legends (page 46-47) have been updated accordingly.
2. We next performed an *in vivo* experiment to examine the role of the fibrotic niche in the colonization of AP cells in the liver (same experiment as mentioned in our response to Reviewer 1's comment 2). We transplanted AKTP-DTR cells to the spleen, and at one week after transplantation, mice were treated with DT for a further week to eliminate AKTP-DTR cells. As suggested by Reviewer 1, we confirmed the remaining fibrotic niche in the liver vessels by Sirius red staining and

α SMA immunohistochemistry. As a secondary transplantation, AP cells were transplanted to the spleen of DT-treated mice, and mouse livers were examined at three to five weeks after transplantation. Notably, we found that AP cells survived and formed a significant number of colonies without AKTP cells. Such tumor lesions were not detected when AP cells were transplanted to control mice without the prior generation of fibrotic niche. These results indicate the role of a fibrotic niche generated by AKTP cells for the survival and colonization of non-metastatic AP cells. We have now added these results as Fig. 5c-f and in the main text (page 13-14). The Methods (page 24) and Figure legends (page 43-44) have been updated accordingly.

Notably, these results appear to conflict with the data in Supplementary Fig. 3, showing that the secondary injection of AP cells into portal vein failed to develop chimeric metastasis by serial seeding. However, we also found by additional experiments that AKTP cells may negatively regulate AP cell proliferation in the liver chimeric tumors (Fig. 3c, f). Thus, it is possible that AP cells formed colonies in this experiment because AKTP cells had been depleted by the DT-DTR system, although confirmation will require a further investigation.

3. TGF- β signaling reportedly induces HSC activation to generate a fibrotic niche for metastasis in the liver (ref. 23, 24). To examine the role of HSCs in liver metastasis, we transplanted metastatic AKTP cells to the spleen of ROSA26-CreER *Tgfbr2* (TGF- β type II receptor gene) flox mice. At one week after transplantation, mice were treated with tamoxifen (Tam) to disrupt the *Tgfbr2* gene, and the tumor phenotype was examined at one to three weeks after Tam treatment. Importantly, the development of metastatic foci was significantly suppressed by blocking the host TGF- β signaling, and α SMA-positive cells were not increased in the stroma. These genetic results support the idea that the fibrotic niche originated by AKTP cell-activated HSCs is required for the generation of metastatic lesions. We have now added these results as Fig. 6c-f and in the main text (page 14-15). The Methods (page 22, 24) and Figure legends (page 45) have been updated accordingly.

We believe that these results strongly support the functional contribution of α SMA-positive HSCs in the development of polyclonal metastasis, including non-metastatic cells. Based on these results, we also added a sentence to the Abstract: "...such a fibrotic microenvironment promotes the colonization of AP cells" (page 2).

3. The data in Fig.5A suggest in fact that HSCs are not sufficient to promote metastatic growth of AP tumor cells. Hence, it is likely that HSCs represent an important component of the niche - but that they are only a marker of advanced lesions or that they are necessary but not sufficient for metastatic growth.

Response: As indicated in our response to comment 2 of Reviewer 2, we showed that the fibrotic niche generated by AKTP cells is important for the colonization of AP cells (Fig. 5c-f). Furthermore, blocking TGF- β signaling via the disruption of *Tgfb2* suppressed the niche generation and tumor formation in the liver (Fig. 6c-f). Moreover, co-culture of AP cells with HSCs in the presence of AKTP-CM or FGF2 significantly increased AP cell proliferation (Fig. 7e, f), although AKTP-DM and FGF2 failed to increase the proliferation of AP cells without HSCs in culture. These results indicate an important role of HSCs in the development of polyclonal metastasis. However, we additionally found that AKTP cell-derived conditioned medium significantly increased the cloning efficiency of AP cells without HSCs (Fig. 7d). Based on these novel results, α SMA-positive HSCs are necessary but may not be solely sufficient for metastatic growth of AP cells, as Reviewer 2 suggested. Accordingly, we have added text describing the concept of a polyclonal metastasis by a driver clone as indirect mechanism through fibrotic niche generation and direct mechanism by secreting factors to the Discussion section (page 17).

4. The co-culture experiments in Fig.5C-D show effects of tumor cell conditioned media on HSC with the strongest effect of AKTP conditioned media, but do not show that this promotes the metastatic growth of tumors. Likewise, the data in Fig.5E-F show interesting phenotypic and gene expression changes upon co-culture, but do not reveal any functions of these in regards to the metastatic process.

Response: In Fig. 6a, b (previous Fig. 5c, d), we showed that conditioned media of AKTP cells activated HSCs. To examine whether or not AKTP-secreted factors

promote polyclonal metastasis, non-metastatic AP cells were co-cultured with HSCs by forming chimeric spheroids and then stimulated with AKTP-CM or one of AKTP cell-secreting factor, FGF2 (same experiment as mentioned in our response to Reviewer 1's comment 9 and Reviewer 2's comment 2). Notably, we found that co-culture with HSCs alone did not increase AP cell proliferation; however, co-culture with HSCs in the presence of stimulation by AKTP-CM or FGF2 significantly increased the Ki67 labeling of AP cells. These results indicate that AKTP-secreted factors including FGF2 play a role in the promotion of polyclonal metastasis by activation of HSCs. The findings are indicated in Fig. 7e, f.

In Fig. 7a, b (previous Fig. 6e, f), we showed that the ES cell signature, including FGF2, was increased in AKTP cells on co-culture with HSCs. As indicated in our response, we found that FGF2 directly increased cloning efficiency of AP cells (Fig. 7d) and that FGF2 indirectly increased proliferation of AP cells by activation of HSCs (Fig. 7e, f). Although FGF2 is one of many factors in AKTP cells upregulated by co-culture with HSCs, these results support the idea that upregulated factors in AKTP cells upon co-culture with HSCs contribute to polyclonal metastasis. We would like to further investigate the precise molecular mechanism underlying polyclonal metastasis through the interaction of AKTP cells and HSCs as our next project.

REVIEWER COMMENTS

Reviewer #1, expert in CRC and liver metastasis (Remarks to the Author):

The authors made a significant effort to address previous concerns and, as a result, the manuscript is much improved.

However, a few issues remain to be addressed to improve clarity:

1) Because the authors highlight the potential clinical implications of their results, the limitations of their model need to be discussed. Namely, in their model, cells are injected into the spleen as means of delivering them to the liver via the portal circulation. Thus, the clinical processes of haematogenous and lymphatic metastasis from primary intestinal tumors are not really fully replicated in this model (e.g. detachment from primary mass, intravasation etc.). These early steps may preferentially select for only highly metastatic, AKTP-like cells for completion of a spontaneous metastasis process. The authors should allude to these potential limitations when discussing the clinical impact of their data.

2) In this regard, when describing the results shown in Supplementary figure 3 the authors state “.....and two weeks later, tdTomato-labeled cells were directly injected into the portal vein, a different route to the liver from the first transplantation (Supplementary Figure 3a)”. This is not really a “different route” as in both cases cells are likely to enter the liver through the portal vein. Also, it is unclear how/why AKTP cells from the second injection incorporate into pre-established micrometastases rather than form independent metastases. This should be discussed.

3) While alpha-SMA is an appropriate marker for activated HSC, it is not specific to these stromal cells.

Authors should discuss other potential stromal cells that may be recruited and contribute to liver colonization.

Fig 5e- The Masson's trichome staining overlaps with cell staining (H&E, AP). What is the source of collagen in these foci? Are the AP cells producing collagen? What is the contribution of Matrigel?

Fig 6 d&e- In the bar graph (d) no significant difference is indicated between the different groups at 2 weeks, yet the images in “e” tell a very different story. Is this an error? Should images of 3 weeks post Tam treatment be shown instead?

Minor point.

Throughout the manuscript the authors use “metastasis” to describe multiple metastatic tumors. This should be replaced with “metastases”. Metastasis is used to describe the process or for a single metastatic tumor.

Reviewer #2, expert in liver fibrosis (Remarks to the Author):

While the revised manuscript is improved a number of major weaknesses still exist:

1. While the Tgfb² deletion experiment is a significant improvement in the revised manuscript, adding some functional data, it is unfortunately not well-designed.

1.a. Tgfb signaling exerts numerous effects and most cells express receptors. Besides fibrogenesis, Tgfb also exerts major immunosuppressive effects, meaning that global Tgfb² deletion can result in strongly increased anti-tumor immunity. Hence, it is not clear whether reduced tumor growth is due to reduced fibrosis or increased anti-tumor immunity.

1.b. The experiment does not address the role of HSC in allowing non-metastatic cell lines to grow.

2. The addition of Fgf2 experiments is interesting. However, the data in Fig.7b showing genes upregulated in AKTP cells co-cultured with HSC need some confirmation as several of these genes are typical HSC genes, such as Des, Gata6, Ncam1 etc, meaning that this could simply be low-grade contamination. In that regard, it is possible that FGF2 is not secreted by AKTP but by HSC. To address this, it would have been important to include FGF2 knockdown of AKTP cells.

3. The manuscript still does not clearly show that HSC, activated by metastatic AKTP cells, allow the growth of non-metastatic cells - this remains speculation as functional experiments addressing this point are missing, and as the appearance of aSMA⁺ cells and fibrosis could just be a marker for its advanced nature.

Point-by-point explanation to the comments by Editor and Reviewers

Our point-by-point responses to the comments made by the Reviewers are included below. The comments are quoted “*verbatim*”, followed by our **responses**. Changes in the manuscript are **highlighted** in the revised text.

Reviewer #1, expert in CRC and liver metastasis (Remarks to the Author):

1) Because the authors highlight the potential clinical implications of their results, the limitations of their model need to be discussed. Namely, in their model, cells are injected into the spleen as means of delivering them to the liver via the portal circulation. Thus, the clinical processes of haematogenous and lymphatic metastasis from primary intestinal tumors are not really fully replicated in this model (e.g. detachment from primary mass, intravasation etc.). These early steps may preferentially select for only highly metastatic, AKTP-like cells for completion of a spontaneous metastasis process. The authors should allude to these potential limitations when discussing the clinical impact of their data.

Response: As the Reviewer pointed out, we used the spleen transplantation model system to examine the process of liver metastasis via portal vein circulation. Accordingly, we have discussed the potential limitations associated with this model when discussing the clinical impact of our data as follows: “However, in the present study, we used a spleen transplantation model to examine the liver metastasis process via portal vein circulation. It thus remains unclear whether or not non-metastatic cells can detach from the primary tumors and intravasate with metastatic cells as polyclonal cell clusters. Clarifying the whole process of polyclonal metastasis will require investigating such early selection steps at the primary site.” (at page 15-16)

2) In this regard, when describing the results shown in Supplementary figure 3 the authors state “and two weeks later, tdTomato-labeled cells were directly injected into the portal vein, a different route to the liver from the first transplantation (Supplementary Figure 3a)”. This is not really a “different route” as in both cases cells are likely to enter the liver through the portal vein. Also, it is unclear how/why AKTP cells from the second injection incorporate into pre-established micrometastases rather than form independent metastases. This should be discussed.

Response: We agree that portal vein injection is not really a “different route” from spleen injection. However, the purpose of this experiment was to avoid possible co-dissemination of secondarily injected cells with initially injected cells from the transplanted site. In response to this comment, we have now revised the sentence explaining the portal vein injection experiment as follows: “tdTomato-labeled cells were directly injected into the portal vein to avoid possible co-dissemination with Venus-AKTP cells from the original transplanted site.” (at page 7)

We apologize for the confusion about the metastases of secondarily injected AKTP cells via the portal vein. We detected liver tumors consisting of only tdTomato-labeled AKTP cells in addition to chimeric tumor foci of tdTomato- and Venus-AKTP cells. These results indicate that AKTP cells can generate metastasis either via clonal dissemination or serial seeding. To clarify this point, we replaced the fluorescent immunohistochemistry photographs in Supplementary Figure 3 top and revised the explanation of the results as follows: “When tdTomato-labeled AKTP cells were injected via the portal vein, we found liver metastases consisting of both tdTomato- and Venus-AKTP cells as well as only tdTomato-AKTP cells (Supplementary Figure 3b *top A* and *B*, respectively), suggesting that AKTP cells can generate metastases either via clonal dissemination or serial seeding.” (at page 7)

3) While alpha-SMA is an appropriate marker for activated HSC, it is not specific to these stromal cells. Authors should discuss other potential stromal cells that may be recruited and contribute to liver colonization.

Response: As shown in Fig. 4d, α SMA-positive cells in the tumor stroma started to proliferate from inside and outside of vessel walls at three days after spleen transplantation. Based on these observations, we suspected that the major origin of these fibrotic cells in the tumor stroma was residential HSCs. However, as the Reviewer pointed out, the α SMA expression is not specific to activated HSCs. There are reportedly three major origins of α SMA-positive myofibroblasts in the liver: HSCs, portal fibroblasts, and bone marrow-derived fibrocytes that proliferate and are activated in liver fibrosis (Nishio et al, J Hepatol, 2019). Therefore, we additionally discussed the potential contribution of other types of stromal cells to the liver colonization of tumor cells as follows: “In this study, we examined the role of HSCs in fibrotic niche generation in liver metastases. However, it has been reported that

α SMA-expressing hepatic myofibroblasts also originate from portal fibroblasts or bone marrow-derived fibrocytes²⁹. Furthermore, the ratio of different-origin-derived α SMA-expressing cells may change during tumor progression, suggesting the possibility that stromal cells originated from portal fibroblasts, or fibrocytes are also recruited in the metastatic foci and contribute to colonization like HSC-originating fibroblast-like cells. However, further investigations will need to be conducted to confirm this point.” (at page 14-15)

Fig 5e- The Masson's trichome staining overlaps with cell staining (H&E, AP). What is the source of collagen in these foci? Are the AP cells producing collagen? What is the contribution of Matrigel?

Response: In the Masson's trichrome staining sections, collagen fiber depositions were colocalized with α SMA-positive cells but not with AP cells (Fig. 5e). Accordingly, it is highly possible that the source of collagen was these fibroblastic cells. Consistently, it has been reported that activated myofibroblasts produce collagen in the liver fibrosis (Iwaisako et al, PNAS, 2014). We also confirmed that Matrigel transplantation did not cause host responses like fibrogenesis in the liver (Fig. 4b bottom). Therefore, we have suggested that the major source of collagen in the tumor stroma might be fibroblastic cells, as follows: “Furthermore, collagen fiber depositions were colocalized with fibroblastic cells in the tumor stroma, suggesting that α SMA-expressing fibroblastic cells were major sources of collagen fibers in these foci (Fig. 5e)” (at page 11)

Fig 6 d&e- In the bar graph (d) no significant difference is indicated between the different groups at 2 weeks, yet the images in “e” tell a very different story. Is this an error? Should images of 3 weeks post Tam treatment be shown instead?

Response: We apologize for not including typical photographs of liver tumors in Fig. 6e. As suggested, we replaced the whole liver images (H&E staining) of no-Tam control and *Tgfb2* KO mice with those at three weeks post-Tam treatment in Fig. 6e. The new photographs correspond to the ratios of the liver tumor areas in the respective models (Fig. 6d).

Minor point.

Throughout the manuscript the authors use “metastasis” to describe multiple metastatic tumors. This should be replaced with “metastases”. Metastasis is used to describe the process or for a single metastatic tumor.

Response: We apologize for the grammatical errors. We have corrected the word “metastasis” to “metastases” as appropriate throughout the text. (highlighted)

Reviewer #2, expert in liver fibrosis (Remarks to the Author):

*1. While the *Tgfb*2 deletion experiment is a significant improvement in the revised manuscript, adding some functional data, it is unfortunately not well-designed.*

*1.a. *Tgfb* signaling exerts numerous effects and most cells express receptors. Besides fibrogenesis, *Tgfb* also exerts major immunosuppressive effects, meaning that global *Tgfb*2 deletion can result in strongly increased anti-tumor immunity. Hence, it is not clear whether reduced tumor growth is due to reduced fibrosis or increased anti-tumor immunity.*

Response: The inhibition of TGF- β signaling by inhibitor treatment reportedly suppressed liver metastasis of mouse CRC cells through anti-tumor immunity by using similar transplantation model (Tauriello et al, Nature, 2018), although the authors of that study did not assess the effect of impaired fibrotic niche generation on TGF- β inhibition-induced growth arrest. Accordingly, as the Reviewer pointed out, we could not discriminate the possible contribution of anti-tumor immunity to the suppression of metastasis in *Tgfb*2 KO mice, although significant suppression of fibrotic niche generation was confirmed (Fig. 6f). As suggested by the Editor, we have discussed the limitations associated with this model and added text regarding the possible contribution of anti-tumor immunity as follows: “In the present study, we showed that fibrotic niche generation was significantly suppressed in *Tgfb*2 knockout mouse liver, which was associated with reduced tumor growth. However, it has also been reported that TGF- β inhibitor treatment suppressed liver metastases of CRC cells through the activation of anti-tumor immunity³⁰. Accordingly, we need to consider the potential contribution of the immune response to cancer cells together with the suppression of niche generation for reducing tumor growth following *Tgfr*2 deletion.” (at page 16)

1.b. The experiment does not address the role of HSC in allowing non-metastatic cell lines to grow.

Response: As the Reviewer noted, we were unable to assess the role of HSCs in allowing non-metastatic AP cells to grow in polyclonal cell clusters using this model because metastatic AKTP cells that lead to polyclonal metastasis did not form metastases in the absence of host TGF- β signaling. However, in the previous revision, we showed that the microenvironment generated by AKTP-DTR cells supported colonization of non-metastatic AP cells in the liver (Fig. 5c-f), and HSCs activated by AKTP-derived conditioned medium or FGF2 increased the proliferation of co-cultured AP cells (Fig. 7f, g). We believe that these results strongly suggest the role of activated HSCs in allowing non-metastatic cells to grow *in vivo*.

2. The addition of Fgf2 experiments is interesting. However, the data in Fig.7b showing genes upregulated in AKTP cells co-cultured with HSC need some confirmation as several of these genes are typical HSC genes, such as Des, Gata6, Ncam1 etc, meaning that this could simply be low-grade contamination. In that regard, it is possible that FGF2 is not secreted by AKTP but by HSC. To address this, it would have been important to include FGF2 knockdown of AKTP cells.

Response: The Reviewer pointed out the possible low-grade contamination of HSCs in AKTP samples for the PCR-Array analysis (Fig. 7b). To exclude such possible cross-contamination, we separately co-cultured AKTP cells and HSCs using Boyden chamber wells and confirmed the induction of FGF2 expression in AKTP cells upon co-culture with HSCs by RT-PCR. We have now added these RT-PCR results to Supplementary Figure 8. (at page 12)

In addition, we confirmed through fluorescent immunohistochemistry that the FGF2 expression was detected in AKTP cells of liver tumors at day 14 after spleen transplantation, while it was not detected at day 3, where the fibrotic niche had not yet been generated (Fig. 4b). These results suggest that HSCs activate AKTP cells to express FGF2. We have now added the fluorescent IHC results to Fig. 7c and a comment as follows: "In addition, we confirmed that the FGF2 expression was induced in AKTP cells in the liver tumor foci at day 14 but not day 3 after spleen transplantation (Fig. 7c), suggesting a role of HSCs in AKTP cell activation." (at page

12) These results support the interaction of HSCs and AKTP cells, as described in page 12.

3. The manuscript still does not clearly show that HSC, activated by metastatic AKTP cells, allow the growth of non-metastatic cells - this remains speculation as functional experiments addressing this point are missing, and as the appearance of α SMA+ cells and fibrosis could just be a marker for its advanced nature.

Response: As we showed in the previous revision, HSCs activated by AKTP cells increased the proliferation of non-metastatic cells by *in vitro* co-culture experiments (Fig. 7f, g). We believe that these data are strong evidence of the role of activated HSCs in allowing the growth of non-metastatic cells, at least during the early step of colonization. The results of *in vivo* experiments in the revision (Fig. 5c-f) also support the role of HSCs in the colonization of non-metastatic cells.

However, non-metastatic AP cells continuously proliferated after the depletion of AKTP cells in the liver (Fig. 3c-f), suggesting that non-metastatic cells require α SMA+ fibroblastic cells only at the early survival and colonization steps but not at the later stages of metastasis, as the Reviewer pointed out.

As suggested by the Editor, we have now discussed this point as a caveat of the present study as follows: "In addition, we showed that once polyclonal metastasis is established, AP cells no longer require AKTP cells for continuous proliferation. Therefore, whether or not activated fibroblastic cells play a role at later proliferation stages of polyclonal metastasis will need to be explored using novel mouse models." (at page 15)